# The RNA-binding protein HuR is required for maintenance of the germinal centre response

Ines C. Osma-Garcia[1], Dunja Capitan-Sobrino[1], Mailys Mouysset[1], Sarah E. Bell [2], Manuel Lebeurrier[1], Martin Turner [2✉] & Manuel D. Diaz-Muñoz [1,2✉]

The germinal centre (GC) is required for the generation of high affinity antibodies and immunological memory. Here we show that the RNA binding protein HuR has an essential function in GC B cells to sustain the GC response. In its absence, the GC reaction and production of high-affinity antibody is severely impaired. Mechanistically, HuR affects the transcriptome qualitatively and quantitatively. The expression and splicing patterns of hundreds of genes are altered in the absence of HuR. Among these genes, HuR is required for the expression of Myc and a Myc-dependent transcriptional program that controls GC B cell proliferation and Ig somatic hypermutation. Additionally, HuR regulates the splicing and abundance of mRNAs required for entry into and transition through the S phase of the cell cycle, and it modulates a gene signature associated with DNA deamination protecting GC B cells from DNA damage and cell death.

[1] Toulouse Institute for Infectious and Inflammatory Diseases (Infinity), Inserm UMR1291, CNRS UMR5051, University Paul Sabatier, CHU Purpan, Toulouse, France. [2] Immunology Program, The Babraham Institute, Babraham Research Campus, Cambridge CB22 3AT, UK. ✉email: martin.turner@babraham.ac.uk; manuel.diaz-munoz@inserm.fr

Germinal centres (GC) are the sites for active B cell proliferation and, following somatic hypermutation (SHM), selection of B cell clones with increased affinity for antigen. GC B cells give rise to long-lived memory B cells and antibody-secreting plasma cells that protect against recurrent infections[1]. Based on cell density, GCs are divided in two main areas: the dark zone (DZ) and the light zone (LZ)[2]. DZ B cells are highly proliferative and undergo immunoglobulin (Ig) SHM[3]. They migrate to the LZ to test their newly mutated B cell receptor (BCR) by capturing antigen from follicular dendritic cells and seeking help from T follicular helper (Tfh) cells[4,5]. The scarcity of antigen makes this a highly competitive process where LZ GC B cells expressing BCRs with the highest affinities are subsequently selected for further rounds of proliferation[6] or for terminal differentiation into plasma cells or memory B cells[7]. Establishment of the GC reaction is stringently regulated by transcriptional and post-transcriptional mechanisms.

To achieve their functions, DZ and LZ GC B cells have very distinctive transcriptomes that reflect the differential expression of transcription factors. Bcl6 is required for naive B cell differentiation into GC B cells upon activation by antigen and Tfh cells[8]. E2A, PAX5 and FOXO1 are all required for proliferation and induction of activation-induced cytidine deaminase (AID)-mediated SHM in DZ B cells[9]. By contrast, activation of nuclear factor (NF)-kB and expression of Myc are required for GC B cell maintenance[10–12]. Myc is mainly expressed by LZ GC B cells expressing high-affinity BCRs that are positively selected by Tfh cells[13]. Myc protein abundance reflects the extent of Tfh cell help received by LZ GC B cells[14] and sets in motion a genetic programme marking further rounds of proliferation and SHM in the DZ[11,12].

Post-transcriptional regulation of gene expression is also an essential regulatory principle of GC responses, but it is far less well understood than the regulation by transcription factors. MicroRNAs have been found to function in the GC B cell by buffering the toxic effects of Myc and AID-mediated DNA damage[15,16]. RNA-binding protein (RBP) regulate cell fate and function by controlling multiple aspects of gene expression, including transcription, processing, subcellular localisation, stability and translation of mRNA into protein[17]. Recent studies have shown that PTBP1, an RBP that is induced by Myc, has an ancillary function in the Myc-dependent gene expression programme of GC B cells[18]. Also, conditional deletion of the RBP human antigen R (HuR; encoded by Elavl1) from the pro-B cell stage onwards has shown its essential effect in B cell activation and antibody production[19,20]. However, the function that HuR plays in the biology of GC B cells remains unknown.

HuR binds to U-rich elements present mostly in introns and 3' untranslated regions (3'UTR) of its RNA targets[21,22]. HuR shuttles from the nucleus to the cytoplasm where it mediates distinct functions. In the nucleus, HuR controls mRNA splicing, whereas in the cytoplasm, it regulates mRNA stability and translation[23]. HuR has been implicated in the control of the cell cycle[24]. Chemical inhibition of HuR translocation to the cytoplasm or blocking of HuR binding to its mRNA targets reduced T-lymphocyte activation and proliferation of different cell lines[25,26]. HuR modulates mRNA translation of Myc in different cell types, including B cell lymphoma cell lines[27–29]. During B cell activation, HuR is required to protect B cells from cell death induced by reactive oxygen species (ROS)[20]. Additionally, HuR may regulate the magnitude of expression of B cell activation markers required for B:Tfh cell interaction during positive selection[19]. Whether these are relevant and contribute to B cell selection in the GC is unknown.

In this work, we show that HuR has a fundamental function in establishing and sustaining the GC B cell responses. HuR is increased in Myc+ LZ GC B cells receiving Tfh cell help, and conditional gene deletion leads to a reduction in Myc expression. HuR is required for the expression of hundreds of genes including those of a Myc-dependent transcriptional programme linked to recycling of antigen-selected LZ GC B cells into the DZ. Additionally, HuR further controls the splicing and expression of mRNA targets required for cell cycle entry and transition through the S phase. In the absence of HuR, antigen-specific class-switched GC B cells are severely impaired showing diminished proliferation and increased DNA damage and GC B cell death. Thus, HuR has an important function in GC B responses allowing selection and expansion of antigen-specific GC B cell clones that confer immunity.

## Results

**GC-dependent humoral responses are impaired in the absence of HuR.** To bypass the activation and metabolic defects leading to the death of follicular HuR knockout (KO) B cells upon cell activation[20] and to study the intrinsic role of HuR in GC B cells, we generated Elavl1^fl/fl Aicda-Cre mice (hereafter named as HuR^fl/fl AID^Cre mice). We used an Aicda-Cre transgenic mouse line expressing a polycistronic mRNA in which the coding sequences of a Cre recombinase and a human CD2 reporter are linked by an internal ribosome entry site (IRES). Analysis of hCD2 and HuR expression in HuR^fl/fl AID^Cre mice showed progressive and efficient gene deletion in GC B cells after establishment of the GC reaction (Supplementary Fig. 1a, b) allowing us to study the intrinsic functions of HuR in GC B cells and GC-mediated humoral responses.

Immunisation of HuR^fl/fl (control) and HuR^fl/fl AID^Cre mice with the thymus-dependent antigen 4-hydroxy-3-nitrophenylacetyl hapten conjugated to Keyhole Limpet Hemocyanin (NP-KLH) in alum followed by analysis of NP-antigen-specific antibodies in the serum of these mice showed that HuR was required for high-affinity, class-switched antibody production. Quantitation of NP23-binding IgM showed no differences between control and HuR^fl/fl AID^Cre mice, while NP23-binding IgG1 was reduced ~5-fold in HuR^fl/fl AID^Cre mice after both primary and secondary immunisation (Fig. 1a). High-affinity NP2-binding IgG1 antibodies increased over time during the immune response in control mice. However, the presence of NP2-binding IgG1 in the serum of HuR^fl/fl AID^Cre mice was negligible and impeded calculation of antibody titres at all the time points analysed (Fig. 1b). Quantitation of the number of antibody-secreting cells (ASCs) producing NP-specific antibodies in the spleen and bone marrow (BM) of these mice showed a 5–10-fold reduction in the number of NP23-binding IgG1 ASCs present in HuR^fl/fl AID^Cre mice. NP2-binding IgG1 ASCs were not detected in HuR^fl/fl AID^Cre mice (Fig. 1c). Thus, HuR is required for the generation of ASCs producing high-affinity and class-switched antibodies.

**HuR is required for the expansion of GCs.** Analysis of GC responses in the spleen and lymph node (LN) of mice immunised with NP-KLH in alum showed that the percentage and number of GC B cells at day 7 post-immunisation was reduced 2-fold in HuR^fl/fl AID^Cre mice compared to control HuR^fl/fl mice or AID^Cre mice (Fig. 2a and Supplementary Fig. 1c–e).

Progression of the GC reaction was severely impaired in HuR^fl/fl AID^Cre mice, with a 7-fold and a 10-fold reduction in the percentage and number of GC B cells compared to control mice at day 10 and day 14 following immunisation (Fig. 2a). By contrast, early establishment of GC responses was not affected in HuR^fl/fl AID^Cre mice as the percentage and number of GC B cells found in the draining LNs of these mice at day 3.5, 4.5 and 5.5 after immunisation were similar to those found in control mice

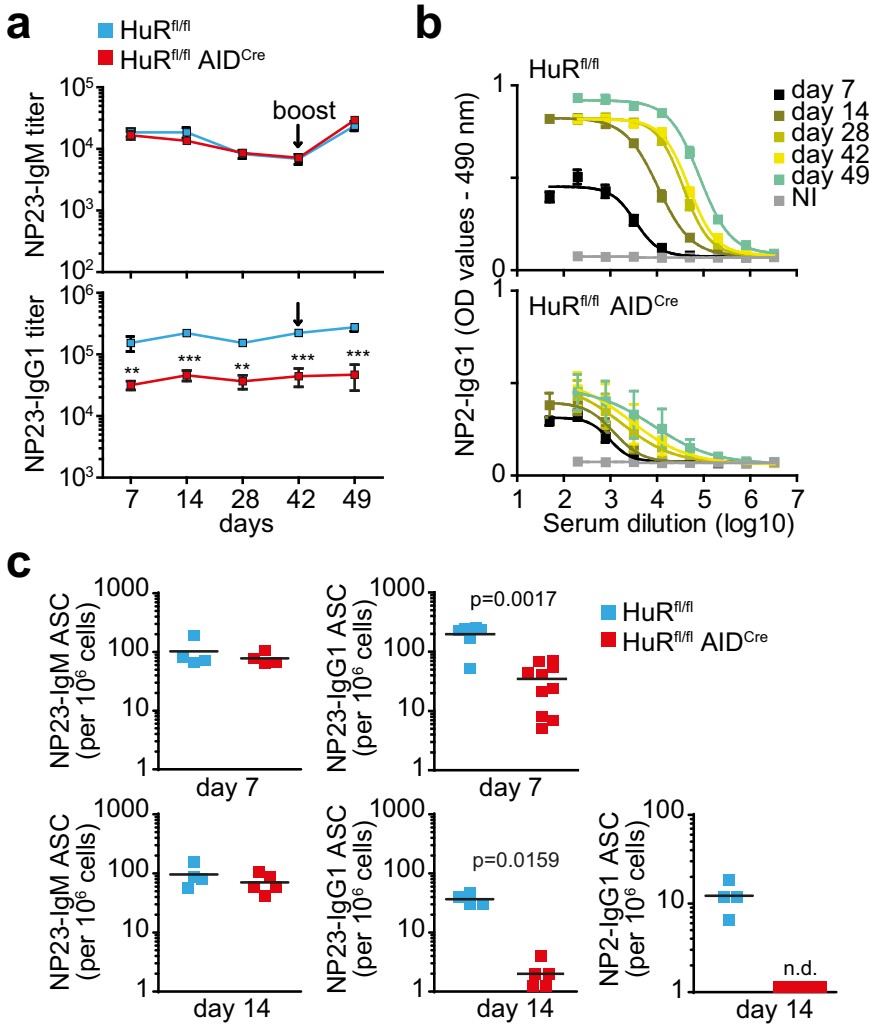

**Fig. 1 Antibody responses are impaired upon deletion of HuR in GC B cells. a** Serum titres of NP-IgM and NP-IgG1 from HuR[fl/fl] and HuR[fl/fl] AID[Cre] mice immunised with NP-KLH in alum (data shown as mean ± SD, $n = 7$ mice per group, two-way ANOVA and Bonferroni's post-test, **$p < 0.01$, ***$p < 0.001$). **b** Serum dilution curves for quantitation of high-affinity NP2-IgG1 in blood serum from mice described in **a** (data shown as mean ± SD, NI = not immunised). **c** Numbers of NP23-IgM, NP23-IgG1 and high-affinity NP2-IgG1-specific antibody-secreting cells (ASCs) in the spleens of HuR[fl/fl] and HuR[fl/fl] AID[Cre] mice at day 7 and day 14 after immunisation (representative data from 1 of 3 independent experiments, $n = 4$–9 mice per group per experiment, two-sided Mann–Whitney test). Source data are provided as a Source data file.

(Supplementary Fig. 1d, e). This was likely a reflection of the time required for transcriptional activation of the AID-Cre transgenic construct and progressive depletion of HuR in GC B cells from HuR[fl/fl] AID[Cre] mice after assembly of GCs (Supplementary Fig. 1a, b). Thus, to assess the requirement of HuR for in vivo B cell activation and development of GCs, we analysed early GC responses in HuR[fl/fl] Mb1[Cre] mice in which deletion of HuR takes place at the pro B cell stage of development. Compared to control mice, the percentage and number of GC B cells found in the draining LNs of HuR[fl/fl] Mb1[Cre] mice were reduced by 2.5-fold and 9-fold at day 4.5 and 5.5 post-immunisation with NP-KLH in alum, respectively (Supplementary Fig. 1d, e). GC responses remained severely impaired in HuR[fl/fl] Mb1[Cre] mice at day 7.5 post-immunisation as previously reported[20]. Taken together, these data suggest that HuR has an important function in the establishment, expansion and/or the maintenance of GCs.

Further analysis of GC responses in HuR[fl/fl] AID[Cre] mice revealed an ~2.5-fold reduction in the percentage and number of total IgG1[+] GC B cells compared to control mice at day 7 after immunisation (Fig. 2b). However the percentage of IgG1[+], IgE[+] and CD138[+] B cells generated after co-culturing naive B cells

with 40LB stroma cells[30] was similar in the presence or absence of HuR (Supplementary Fig. 2a). This indicates that HuR is likely dispensable for Ig class-switch recombination (CSR) and terminal differentiation into plasma cells.

Further characterisation of GC B cells based on cell surface expression of markers of DZ and LZ GC cells revealed a reduction in the percentage of DZ GC B cells and a corresponding increase of LZ GC B cells in HuR[fl/fl] AID[Cre] mice compared to control mice at day 7 and day 14 following immunisation (Fig. 2c and Supplementary Fig. 2b). Visualisation by confocal microscopy of GCs in the spleen of immunised mice at day 7 confirmed a reduction in GC size and the altered GC cell distribution between the DZ and the LZ in the absence of HuR (Fig. 2d).

The proportion of apoptotic GC B cells labelled ex vivo with a viability dye for flow cytometry, and/or with VAD-FMK that recognise active caspases, was increased between 1.5- and 3-fold in HuR[fl/fl] AID[Cre] mice compared to control mice (Fig. 2e and Supplementary Fig. 2c). This was not a consequence of altered mitochondrial metabolism as the analysis of mitochondrial mass, mitochondrial potential or ROS levels showed no changes in GC B cells from HuR[fl/fl] AID[Cre] mice compared to control mice

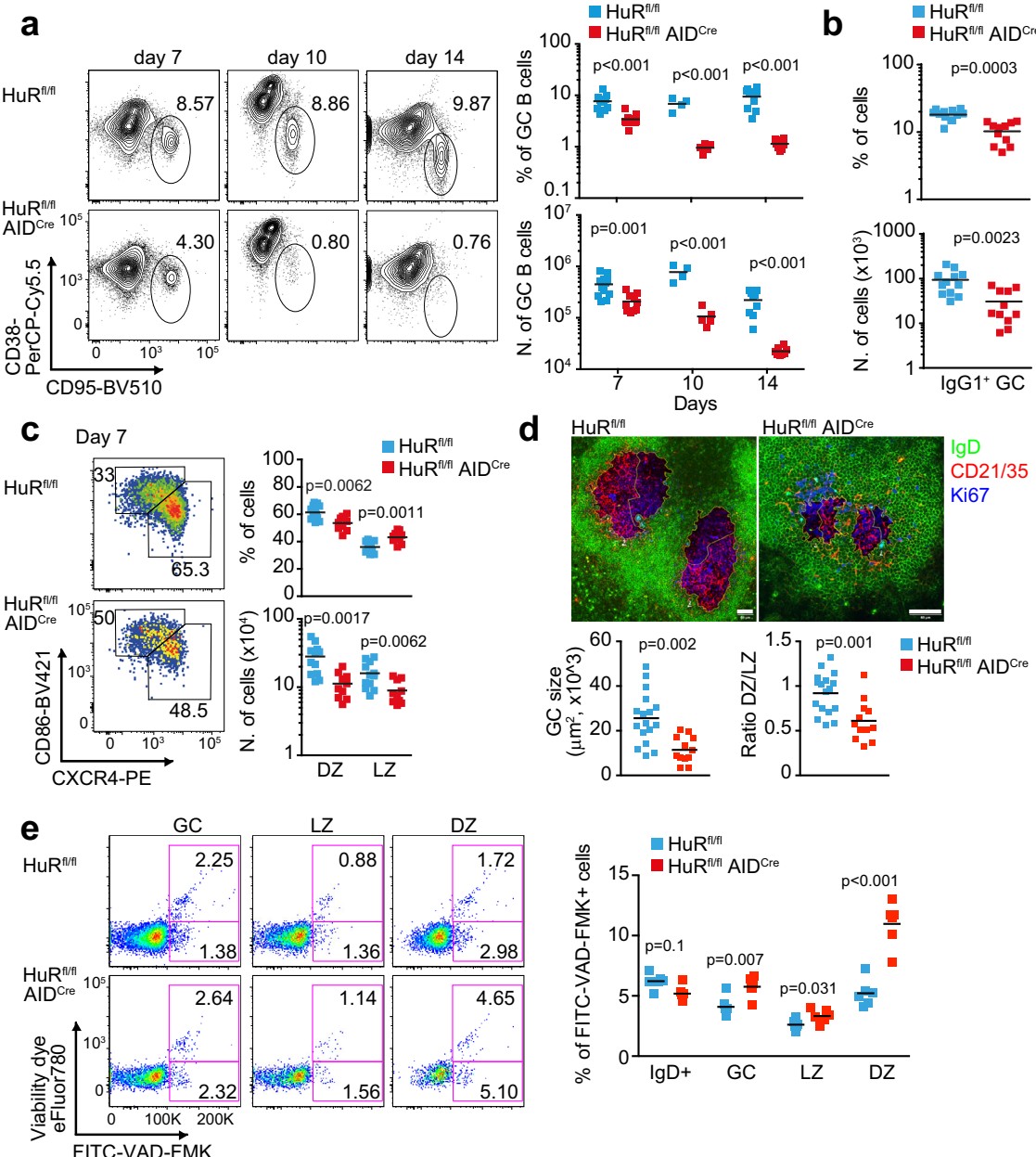

**Fig. 2 HuR is essential for maintenance of the GC reaction. a** Flow cytometric analysis of GC B cells in the spleen of HuR$^{fl/fl}$ and HuR$^{fl/fl}$ AID$^{Cre}$ mice at day 7, day 10 and day 14 after immunisation with NP-KLH in alum. Left panels, representative contour plots of CD95$^+$ CD38$^-$ GC B cells (previously gated on CD19$^+$ IgD$^-$ cells). Right panels, the percentage and number of GC B cells (data representative from 1 of at least 3 independent experiments performed at each time point, $n = 4$–12 mice per group depending on the day of the analysis, two-sided Mann–Whitney test). **b** Percentage and number of IgG1$^+$ GC B cells in HuR$^{fl/fl}$ and HuR$^{fl/fl}$ AID$^{Cre}$ mice at day 7 after immunisation (GC B cells gated on CD19$^+$ IgD$^-$ CD38$^-$ CD95$^+$). **c** Analysis of DZ and LZ GC B cells at day 7 after immunisation in mice shown in **b**. Left panels, representative contour plots of CXCR4$^+$ CD86$^-$ DZ GC B cells and CXCR4$^-$ CD86$^+$ LZ GC B cells (previously gated on CD19$^+$ IgD$^-$ CD38$^-$ CD95$^+$). Right panels, quantitation of the percentage and number of DZ and LZ GC B cells Data in **b**, **c** are from 2 independent experiments, $n = 12$ (HuR$^{fl/fl}$) and $n = 11$ (HuR$^{fl/fl}$ AID$^{Cre}$) mice, two-sided Mann–Whitney test. **d** Visualisation by confocal microscopy of GC size and GC B cell distribution in HuR$^{fl/fl}$ and HuR$^{fl/fl}$ AID$^{Cre}$ mice at day 7 after immunisation (scale bar = 50 μm). Bottom panels, quantitation of the GC size and the ratio DZ/LZ quantified upon staining of the DZ with Ki67 (in blue) and the LZ with CD21/35 (in red). $n = 17$ (HuR$^{fl/fl}$) or $n = 12$ (HuR$^{fl/fl}$ AID$^{Cre}$) GCs examined from 2 immunised mice per genotype over 2 independent experiments (two-sided Mann–Whitney test). **e** Representative pseudo-colour plots showing the staining with the viability dye eFluor780 and FITC-VAD-FMK of the different GC B cell populations in the spleen of HuR$^{fl/fl}$ and HuR$^{fl/fl}$ AID$^{Cre}$ mice at day 10 after immunisation with NP-KLH alum. Right panel, quantitation of the percentage of B cells positive for FITC-VAD-FMK ($n = 6$ mice per group, 2-way ANOVA and Bonferroni's post-test). Source data are provided as a Source data file.

(Supplementary Fig. 2d). DZ GC B cells showed the highest proportion of non-viable cells. The percentage of DZ GC B cells labelled with FITC-VAD-FMK and/or the viability dye was significantly increased in the absence of HuR at all times analysed following immunisation. By contrast, the percentage of non-

viable LZ GC B cells was similar in HuR$^{fl/fl}$ AID$^{Cre}$ mice compared to control mice at day 7 after immunisation (Supplementary Fig. 2c), but it was increased 1.4-fold at day 10 (Fig. 2e) and 5-fold at day 14 in HuR$^{fl/fl}$ AID$^{Cre}$ mice (Supplementary Fig. 2c). Taken together, HuR is required for

the survival of both LZ and DZ GC B cells and maintenance of GCs over time.

**HuR is needed for NP-specific GC B cell selection and expansion.** The increased death of LZ GC B cells in HuR$^{fl/fl}$ AID$^{Cre}$ mice suggested a role for HuR during selection and expansion of NP-specific B cell clones. Consistent with this, analysis of the percentage of NP$^+$ GC B cells showed a 1.6-fold reduction in HuR$^{fl/fl}$ AID$^{Cre}$ mice compared to control mice at day 10 after immunisation (Fig. 3a). The percentage of IgG1-class-switched NP$^+$ GC B cells was reduced even further with a 7-fold reduction in HuR$^{fl/fl}$ AID$^{Cre}$ mice compared to control mice. By contrast, the percentage of IgG1-class-switched NP$^-$ GC B cells was unaltered, whereas the percentage of non-class-switched NP$^-$ GC B cells was significantly increased in HuR$^{fl/fl}$ AID$^{Cre}$ mice (Fig. 3a and Supplementary Fig. 3a). Importantly, analysis of the appearance of NP$^+$ IgG1$^+$ B cells in early GCs formed at days 3.5 and 5.5 did not detect any significant differences in HuR$^{fl/fl}$ AID$^{Cre}$ mice compared to control mice (Supplementary Fig. 3b, c) suggesting that HuR is required primordially for the expansion of antigen-specific class-switched B cells in GCs.

HuR was efficiently deleted in over 90% of GC B cells from HuR$^{fl/fl}$ AID$^{Cre}$ mice at day 7 after immunisation (Fig. 3b). However, only ~70% GC B cells from HuR$^{fl/fl}$ AID$^{Cre}$ mice lacked the expression of HuR at day 14, suggesting a depletion of HuR-insufficient B cells as the GC reaction progresses. To interrogate whether HuR deletion confers a competitive disadvantage to GC B cells, we reconstituted the immune system of Rag2 KO mice by transferring BM from wild type (CD45.1$^+$) and HuR$^{fl/fl}$ (HuR$^+$ CD45.2$^+$) or HuR$^{fl/fl}$ AID$^{Cre}$ (HuR$^-$ CD45.2$^+$) mice in a ratio of 1:3 (Supplementary Fig. 3d). Analysis of GC responses in these chimeric mice at day 7 and day 10 after immunisation with NP-KLH showed a progressive reduction in the percentage of HuR$^-$ CD45.2$^+$ GC B cells and a concomitant increase of HuR$^+$ CD45.1$^+$ GC B cells over time (Fig. 3c). Analysis of the distribution of GC B cells in these chimeric mice showed a 2-fold increase in the percentage of HuR$^-$ CD45.2$^+$ GC B cells into the LZ compared to controls (Fig. 3d) reinforcing the evidence that HuR expression was required for B cell distribution within GCs. Additionally, we observed that antigen-specific class-switched HuR$^-$ GC B cells were highly reduced with NP$^+$ IgG1$^+$ HuR$^-$ CD45.2$^+$ B cells representing only a 10% of the total number of NP$^+$ IgG1$^+$ B cells present in GCs at day 10 (Fig. 3e). This was a 7-fold reduction in the percentage of NP$^+$ IgG1$^+$ CD45.2$^+$ GC B cells found in mice reconstituted with control cells and prompted us to investigate whether selection of antigen-specific GC B cells was affected in the absence of HuR. To this end, we crossed HuR$^{fl/fl}$ AID$^{Cre}$ mice with a $Myc^{GFP/GFP}$ (GFP-Myc) reporter mouse[31]. Quantitation of GFP-Myc$^+$ GC B cells revealed a 1.5-fold reduction in percentage and a 2-fold reduction in cell number in HuR$^{fl/fl}$ Myc$^{GFP/GFP}$ AID$^{Cre}$ compared to control mice (Fig. 3f). Taken together, our data indicate that HuR expression is required for the selection, expansion and/or maintenance of antigen-specific GC B cells and modulates their distribution within the DZ and LZ compartments.

**HuR expression is increased in positively selected GC B cells.** To define more precisely how HuR is involved in the selection of GC B cells, we sought to understand further the regulation of HuR in GC B cells. To this end, we reanalysed previously published RNAseq data sets that identified three different subsets of LZ GC B cells with different BCR affinity and capacity to establish stable contacts with Tfh cells[7]. $Elavl1$ mRNA (which encodes for HuR) was significantly increased 4-fold in Bcl6$^{hi}$ CD69$^{hi}$ LZ GC B cells with lower-affinity BCRs and expressing high levels of $Myc$ and $Tfap4$ mRNA (Fig. 4a and Supplementary Fig. 4a). Similarly, analysis of the transcriptome of sorted Myc$^+$ AP4$^+$ LZ GC B cells[32] showed a 2-fold increase in the expression of $Elavl1$ mRNA when compared to Myc$^-$ AP4$^-$ LZ GC B cells (Supplementary Fig. 4b). These data suggest that $HuR$ mRNA increases in LZ GC B cells following T cell help.

Consistent with this, we found $Elavl1$ mRNA to be increased in B cells co-cultured with Tfh cells compared to naive or GC B cells (Fig. 4b). $Myc$ and $Tfap4$ mRNAs were also increased in these B cells indicating that they were efficiently receiving help from Tfh cells. Importantly, blocking help provided by Tfh cells by adding Tfr cells into the co-cultures prevented the induction of $Elavl1$, $Myc$ and $Tfap4$ mRNA. Activation through the CD40 receptor and the BCR rapidly increased the expression of HuR in B cells in vitro (Supplementary Fig. 4c). Thus, to confirm that HuR protein expression was augmented in LZ GC B cells that received a positive selection signal, we immunised Myc$^{GFP/GFP}$ reporter mice with NP-KLH in alum. HuR protein abundance was similar between DZ and LZ GC B cells (Fig. 4c) but increased 2-fold in GFP-Myc$^+$ GC B cells compared to total GC B cells (Fig. 4c) indicating that HuR is induced in GC B cells in response to T cell help.

**HuR binds to Myc mRNA and regulates Myc protein expression.** Myc protein abundance in LZ GC B cells is proportional to the strength of help received from Tfh cells and it controls growth and division destiny of LZ GC B cells after entry into the DZ[14]. In cultured cell lines, HuR has been implicated in the post-transcriptional regulation of Myc[27–29,33] prompting the hypothesis that HuR controls Myc expression in GC B cells. In support of this, we sought evidence of a direct interaction between HuR and $Myc$ mRNA by performing individual-nucleotide cross-linking immunoprecipitation (iCLIP) in primary B cells. Analysis of HuR:RNA interactions showed that HuR was bound to U-rich elements present in the 3'UTR of $Myc$ in both resting and mitogen-activated B cells (Fig. 4d). This suggested that HuR might be required for the induction of $Myc$ upon GC B cell positive selection by Tfh cells. Indeed, quantification of Myc protein in positively selected GC B cells from HuR$^{fl/fl}$ Myc$^{GFP/GFP}$ AID$^{Cre}$ and HuR$^{fl/fl}$ Myc$^{GFP/GFP}$ mice revealed a significant reduction in the expression of Myc in the absence of HuR (Fig. 4e). Additionally, HuR expression was required for the complete induction of Myc upon in vitro activation of B cells with mitogens (Fig. 4f). This indicates that HuR plays a direct role in the induction of Myc expression in B cells even in the absence of Tfh cell help.

To establish the importance of reduced Myc expression to the phenotype of HuR$^{fl/fl}$ AID$^{Cre}$ mice, we introduced the MYC-IRES-CD2 transgene that is expressed constitutively from the Rosa26 locus[11]. Overexpression of Myc in HuR KO GC B cells did not rescue GC cell numbers, cell distribution in DZ/LZ or generation of class-switched GC B cells (Supplementary Fig. 4d). Thus, HuR is increased in positively selected LZ GC B cells and contributes to the expression of Myc, but HuR has additional functions in GC B cells that cannot be compensated by enforced expression of Myc.

**HuR controls RNA splicing and abundance in DZ and LZ GC B cells.** To gather insight into the molecular and cellular mechanisms controlled by HuR in GCs, we sorted DZ and LZ GC B cells from HuR$^{fl/fl}$ (control) and HuR$^{fl/fl}$ AID$^{Cre}$ mice immunised with NP-KLH alum for 7 days to analyse their transcriptome by mRNAseq. Principal component analysis of transcriptomics data showed sample clustering based on DZ and

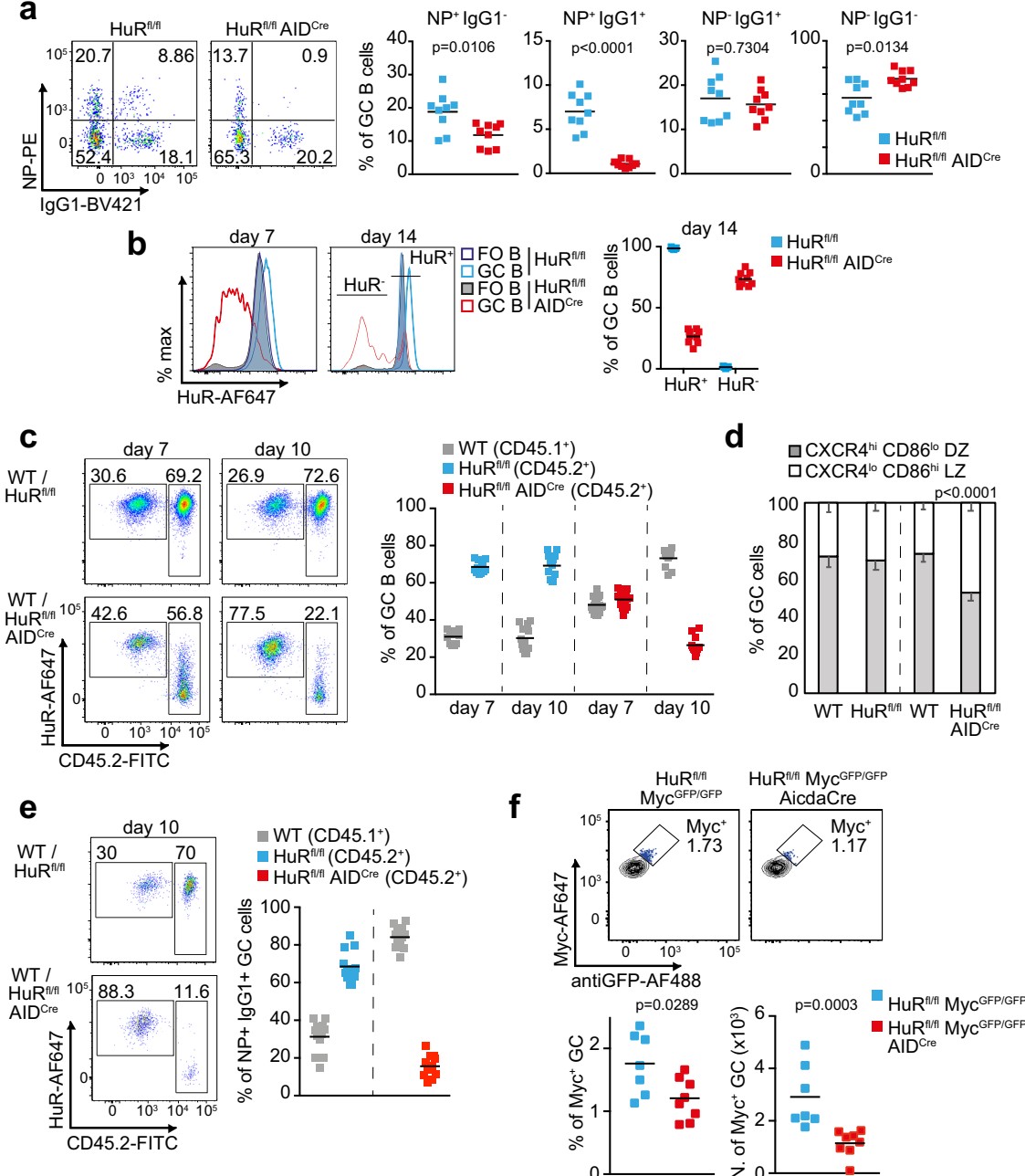

**Fig. 3 HuR is required for the expansion of antigen-specific GC B cells. a** Pseudo-colour plot of NP-specific GC B cells in the spleen of HuR$^{fl/fl}$ and HuR$^{fl/fl}$ AID$^{Cre}$ mice at day 10 after immunisation with NP-KLH in alum. Right panels show the percentage of GC B cells based on surface expression or not of IgG1 and binding of NP antigen. Data are representative from 1 of the 3 independent experiments performed, $n = 9$ mice per group and experiment, two-sided Mann–Whitney test). **b** Histograms showing HuR expression in splenic FO and GC B cells from HuR$^{fl/fl}$ and HuR$^{fl/fl}$ AID$^{Cre}$ mice at day 7 and day 14 after immunisation. Right panel, the percentage of HuR$^+$ and HuR$^-$ GC B cells in HuR$^{fl/fl}$ and HuR$^{fl/fl}$ AID$^{Cre}$ mice at day 14 after immunisation (data representative from 1 of the 3 independent experiments performed ($n = 9$ mice per group and experiment)). **c** Pseudo-colour plots and analysis of the proportion of CD45.2$^-$ and CD45.2$^+$ GC B cells in draining LNs of BM chimera mice (WT CD45.1$^+$/HuR$^{fl/fl}$ CD45.2$^+$ or CD45.1$^+$/HuR$^{fl/fl}$ AID$^{Cre}$ CD45.2$^+$, ratio 1:3) after immunisation with NP-KLH in alum. **d** Percentage of WT (CD45.2$^-$) GC B cells and HuR$^{fl/fl}$ or HuR$^{fl/fl}$ AID$^{Cre}$ (CD45.2$^+$) GC B cells expressing the cell surface markers of DZ and LZ GC B cells in BM chimera mice. Data are presented as mean values $+/-$ SEM. **e** Analysis of the proportion of CD45.2$^-$ and CD45.2$^+$ NP-specific IgG1$^+$ GC B cells in mice shown in **c**. Data in **c–e** are from $n = 12$ or 16 mice per group depending on the genotype and the day of analysis. Data in **e** are from $n = 12$ mice per group. Two-sided Mann–Whitney test in **d**. **f** Myc$^+$ GC B cells in draining LNs of HuR$^{fl/fl}$ Myc$^{GFP/GFP}$ and HuR$^{fl/fl}$ Myc$^{GFP/GFP}$ AID$^{Cre}$ mice at day 7 after immunisation with NPKLH in alum (s.c.). Top panels, representative contour plots of Myc staining and GFP detection (previously gated on CD19$^+$ IgD$^-$ CD95$^+$ CD38$^-$ cells). Bottom panels, percentage and number of Myc$^+$ GC B cells ($n = 7$ (HuR$^{fl/fl}$) or $n = 8$ (HuR$^{fl/fl}$ AID$^{Cre}$) mice, two-sided Mann–Whitney test). Source data are provided as a Source data file.

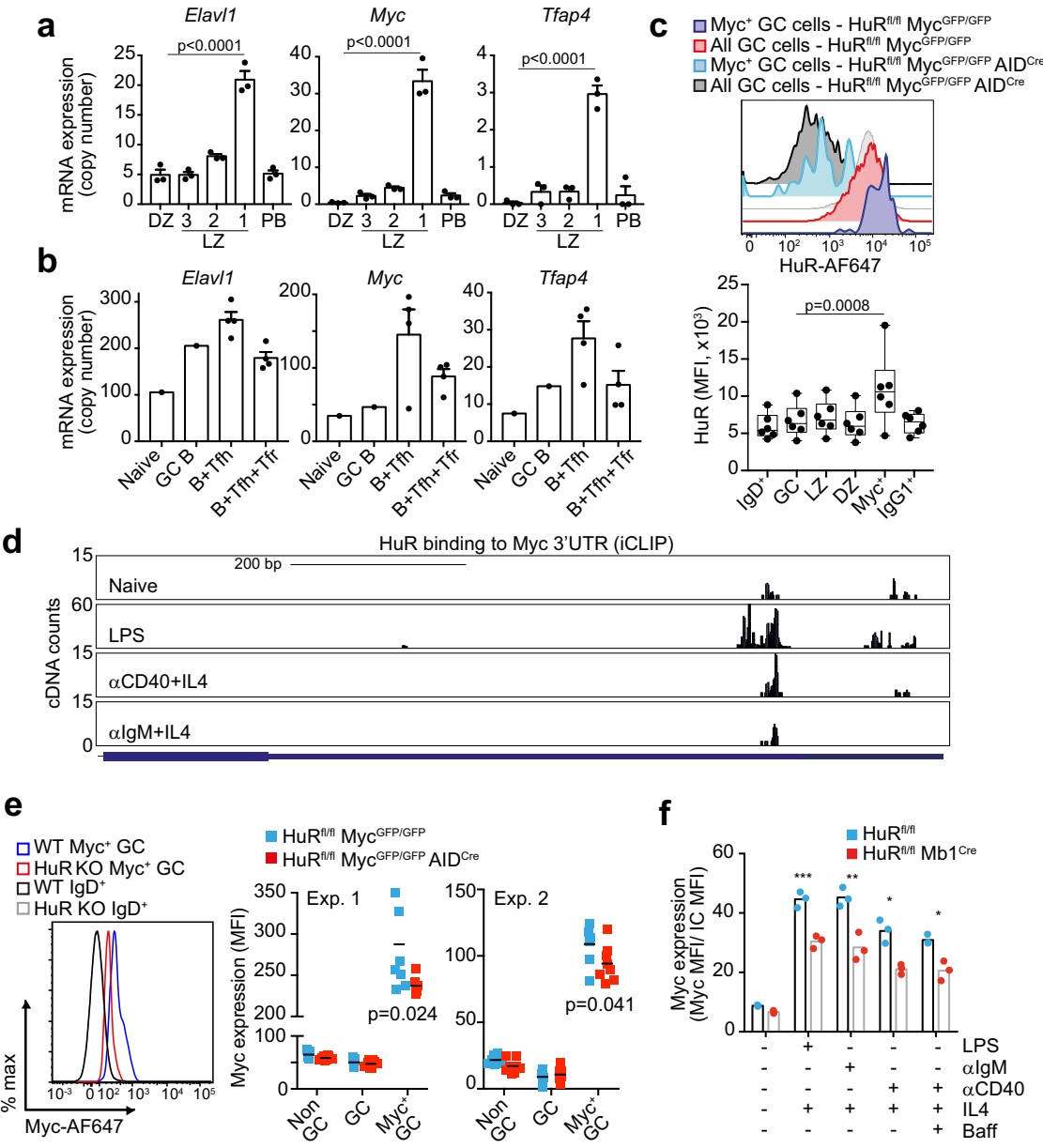

**Fig. 4 HuR regulates Myc expression. a** *Elavl1*, *Myc* and *Tfap4* mRNA expression in DZ B cells and in NP[+] IgG1[+] LZ GC B cells (grouped by the amount of help received from Tfh cells as Fr. 1, Bcl6[lo] CD69[hi]; Fr. 2, Bcl6[hi]CD69[hi]; and Fr. 3, Bcl6[hi]CD69[lo]) and in splenic plasmablasts (PB, NP[+] IgG1[+] CD138[hi]) (GSE109732[7], $n = 3$ samples per group, mean ± SD, DESeq2 analysis with BH correction of $p$ values). **b** *Elavl1*, *Myc* and *Tfap4* mRNA expression in FO and GC B cells and in B cells co-cultured with Tfh cells or Tfh and Tfr (T follicular regulatory) cells (GSE82003[58], $n = 1$ (naive and GC B cells) or $n = 4$ (B+Tfh cells and B+Tfh+Tfr cells), mean ± SD). **c** HuR protein expression in Myc[+] GC B cells. Top panel, histogram showing HuR staining in all GC B cells or in Myc[+] GC B cells from HuR[fl/fl] Myc[GFP/GFP] and HuR[fl/fl] Myc[GFP/GFP] AID[Cre] mice at day 7 after immunisation. Bottom panel, mean fluorescence intensity (MFI) of HuR in naive IgD[+], GC, LZ, DZ, Myc[+] GC and IgG1[+] GC B cells (representative data from 1 of the 2 independent experiments, $n = 6$ mice, data presented as box plots showing the mean, 25–75% range and whiskers from Min. to Max., two-way ANOVA and Tukey's post-test). **d** Analysis by iCLIP of in vivo binding of HuR to the 3'UTR of *Myc* in follicular B cells treated or not with LPS, αCD40+IL4 or αIgM+IL4 for 48 h. **e** Myc protein expression analysis by flow cytometry in GC B cells. Left panel shows a histogram of Myc expression in positively selected GC B cells (gated as shown in Fig. 3f) from HuR[fl/fl] Myc[GFP/GFP] and HuR[fl/fl] Myc[GFP/GFP] AID[Cre] mice at day 8 upon immunisation with NP-KLH in alum. Right panel, quantitation of the mean fluorescence intensity of Myc in GFP[+] GC B cells (data of the 2 independent experiments performed is presented, $n = 7$ or $n = 8$ per genotype depending on the experiment, two-sided Mann–Whitney test). **f** Myc protein expression in follicular B cells from HuR[fl/fl] and HuR[fl/fl] Mb1[Cre] mice activated in vitro for 3 h with the mitogens and cytokines indicated (data representative from 1 of the 3 independent experiments performed, $n = 3$ per group, two-way ANOVA and Tukey's post-test, *$p < 0.05$, **$p < 0.01$, ***$p < 0.001$). Source data are provided as a Source data file.

LZ GC cell type, but not on genotype (Supplementary Fig. 5a). We confirmed this was not due to inefficient Cre-mediated deletion of HuR in HuR[fl/fl] AID[Cre] GC B cells as we did not detect reads mapped to the floxed exon 2 of the *Elavl1* gene in libraries from HuR KO cells (Supplementary Fig. 5b). Thus, we conclude that

HuR deficiency does not globally alter the transcriptome of GC B cells but might affect mRNA abundance in a gene-specific manner.

Differential gene expression analysis showed that the expression of >2000 genes changed between DZ and LZ GC B cells (false discovery rate (FDR) < 0.05) (Fig. 5a and Supplementary Data 1).

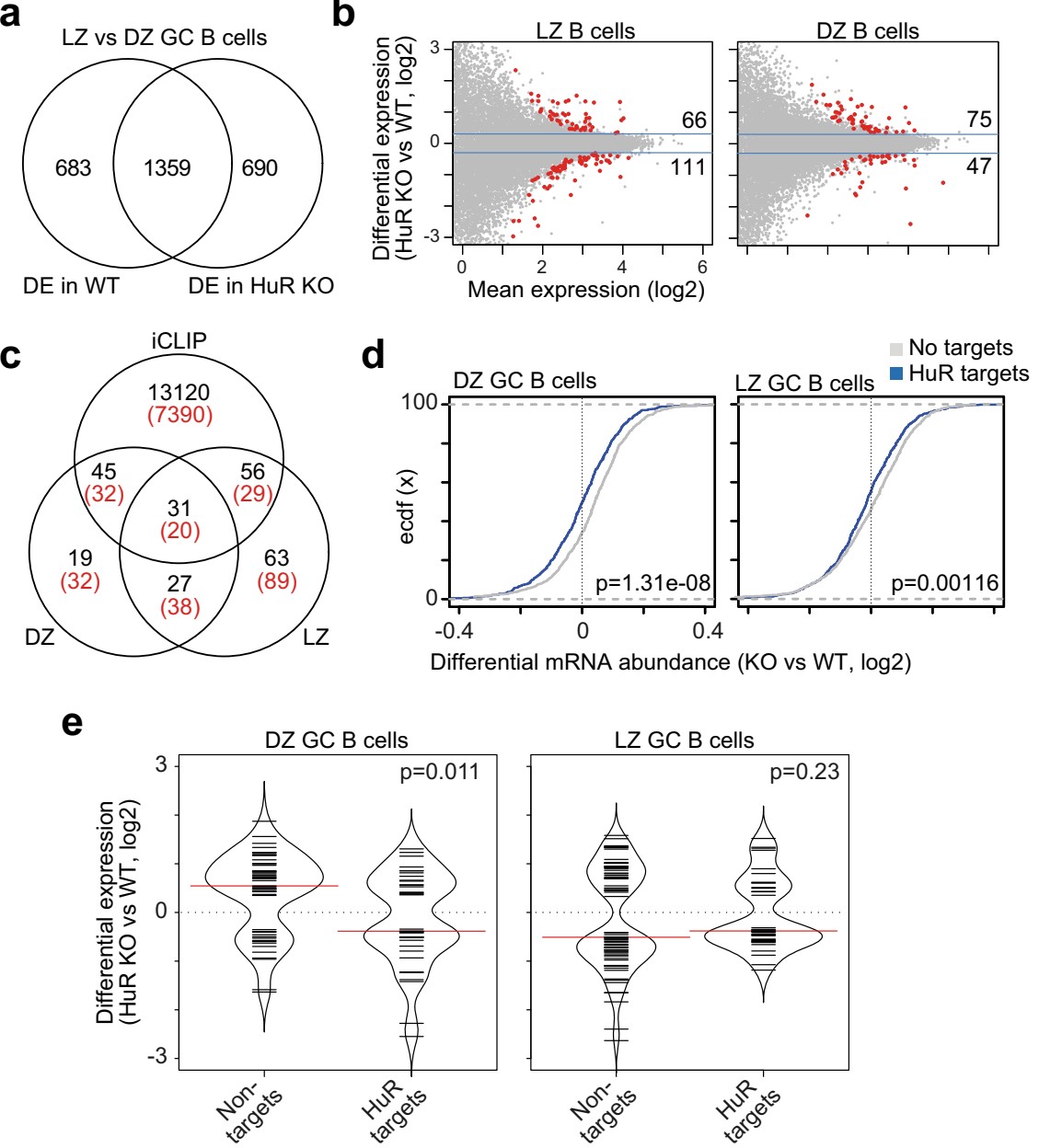

**Fig. 5 Transcriptomics identity of DZ and LZ GC B cells is controlled by HuR. a** Venn diagram showing the overlap of genes that are differentially expressed (DE) between LZ and DZ GC B cells in HuR^fl/fl (control = WT) and HuR^fl/fl AID^Cre (HuR KO) mice (DESeq2, BH correction, FDR < 0.05). **b** MA plots with DE genes identified in HuR KO LZ and HuR KO DZ GC B cells compared to control cells. **c** Venn diagram showing the overlap of DE genes in HuR KO LZ, HuR KO DZ GC B cells and genes of which mRNA transcripts were identified as targets of HuR by iCLIP (in black—number of genes identified as targets of HuR in LPS-stimulated B cells and with at least one significant crosslink, BH correction, FDR < 0.05; in red—genes identified as HuR targets in all conditions of B cells treated with LPS, αCD40+IL4 or αIgM+IL4). **d** Cumulative distribution of differential mRNA abundance in DZ KO and LZ GC KO GC B cells compared to WT cells of HuR mRNA targets and mRNAs not bound by HuR. All mRNA transcripts were merged by gene and subsequently subdivided into targets of HuR or not upon integration of HuR iCLIP data from LPS-stimulated B cells (HuR binding defined as annotation of at least five unique cDNA counts per x-link site in the 3'UTR). Data are from the top 25% of most expressed genes (Kolmogorov–Smirnov test). **e** Bean plot of DE genes in HuR KO DZ GC B cells (left panel) and HuR KO LZ GC B cells (right panel) compared to control WT cells and clustered based on the binding of HuR to their RNA transcripts (minimum 5 unique cDNA counts annotated per x-link site by iCLIP and FDR < 0.05, Kolmogorov–Smirnov test).

A total of 1359 genes were differentially expressed (DE) between DZ and LZ GC B cells independently of the genotype. Additionally, the expression of 683 genes changed significantly between DZ and LZ GC B cells only in control mice, whereas other 690 genes were DE only when comparing DZ and LZ GC B cells from HuR^fl/fl AID^Cre mice (Fig. 5a and Supplementary Data 1), suggesting that HuR modulates the magnitude of changes in mRNA abundance between DZ and LZ GC B cells.

Further comparison of control and HuR KO GC B cells identified changes in the expression of 177 genes in LZ GC B cells and 122 genes in DZ GC B cells in the absence of HuR (Fig. 5b and Supplementary Data 1). 87 out of the 177 genes and 76 out of the 122 genes encoded for mRNAs identified as targets of HuR in our HuR:RNA interactome (iCLIP) analysis performed in lipopolysaccharide (LPS)-activated B cells[20] (Fig. 5c). Importantly, up to 70% of these mRNA transcripts were also identified

as targets of HuR in B cells stimulated through the BCR or the CD40 receptor (Fig. 5c and Supplementary Fig. 5c), even with a variable depth in the different iCLIP data sets (Supplementary Fig. 5d). Further analyses showed that the proportion of HuR mRNA targets was not increased among those genes that were DE in HuR KO GC B cells compared to all genes. Thus, we concluded that HuR selectively controls the abundance of hundreds of mRNAs in DZ and LZ GC B cells.

HuR recognition of RNA regulatory elements in 3'UTRs has been associated with mRNA stabilisation and altered translation into protein. To assess whether the directionality of changes in gene expression in HuR KO GC B cells was associated with the binding of HuR, we grouped genes based on whether their mRNA transcripts were found associated with HuR in the 3'UTR in our HuR iCLIP data. Empirical cumulative distribution function analysis showed that HuR binding to 3'UTRs correlated with an increase in mRNA abundance of its targets in DZ and LZ GC B cells, particularly of those that were more abundantly expressed (Fig. 5d and Supplementary Fig. 5e, f). Similarly, 21 out of 39 DE genes of which mRNA transcripts were bound by HuR were less abundant in HuR KO DZ GC B cells compared to control DZ GC cells. By contrast, most DE genes that were not targeted by HuR (36 out of 51) were increased in HuR KO DZ GC B cells (Fig. 5e). We did not observe any differences in the direction of changes in gene expression when LZ GC B cells were analysed (Fig. 5e). This was likely due to the expression of most genes being diminished in HuR KO LZ GC B cells compared to control LZ GC B cells, and/or the fact that our iCLIP data from mitogen-activated B cells did not fully recapitulate the transcriptome of LZ GC B cells. In summary, we observed that HuR is required to maintain mRNA abundance in DZ and LZ GC B cells.

RNA splicing controls the production of alternative mRNA transcripts, and if dysfunctional, it can promote RNA degradation by non-sense-mediated mRNA decay. Thus, we assessed the impact of the loss of HuR on alternative mRNA splicing (AS) in GC B cells. Comparison of the transcriptome of DZ and LZ WT GC B cells showed 333 significant AS events affecting to 310 genes (rMATS, FDR < 0.05 and absolute changes in inclusion levels >10%) (Fig. 6a and Supplementary Data 2). In the absence of HuR, the number of AS events rose by 1.8-fold in DZ GC B cells and by 3-fold in LZ GC B cells (Fig. 6a and Supplementary Data 2). A total of 595 splicing events affecting 489 genes were differentially regulated in DZ KO GC B cells compared to control DZ GC B cells. By contrast, 991 AS events affecting 806 genes were identified in LZ KO GC B cells. Of these AS events, 208 events associated with 154 genes were commonly found in DZ and LZ GC B cells (Fig. 6b). This was just 13% of all 1586 AS events that were detected in HuR KO DZ and LZ GC B cells, which suggested that HuR was involved in mRNA editing programmes that were cell-type-dependent. Integration of our RNA splicing analysis with HuR iCLIP data revealed that >75% of AS genes in DZ and LZ GC B cells were bound by HuR, a fold enrichment compared to the total number of genes targeted by HuR at introns (Fig. 6c and Supplementary Fig. 5g). However, AS was not associated with changes in mRNA abundance in the absence of HuR as only 7 and 14 genes were both alternatively spliced and DE in HuR KO DZ and LZ GC B cells. Taken together, our data show that HuR is an important splicing modulator in DZ and LZ GC B cells.

**HuR controls pathways linked to cell cycle and DNA damage in GC B cells**. Next, we assessed the biological consequences of the identified changes in splicing and transcript abundance in HuR KO GC B cells. Gene set enrichment analysis (GSEA; Fig. 6d and Supplementary Data 3) showed that changes in splicing in HuR

KO DZ B cells associated with RNA and DNA metabolic processes linked to chromatin organisation (FDR = $1.16 \times 10^{-4}$), mRNA processing (FDR = $3.71 \times 10^{-4}$), cell cycle (FDR = $7.91 \times 10^{-3}$) and DNA recombination (FDR = $2.19 \times 10^{-2}$). In HuR KO LZ GC B cells, changes in splicing were linked to the same biological functions. We also identified gene signatures associated with cellular responses to DNA damage (FDR = $9.04 \times 10^{-13}$), signal transduction by p53 (FDR = $4.70 \times 10^{-8}$), cell cycle (FDR = $4.70 \times 10^{-8}$) and DNA repair (FDR = $6.42 \times 10^{-8}$). Thus, HuR modulates splicing programmes in GC B cells associated with GC B cell proliferation and the response to DNA damage.

GSEA showed that genes with altered expression in the absence of HuR were associated with similar gene sets to the ones linked with changes in splicing (Fig. 6e and Supplementary Data 3). Genes that were DE between DZ and LZ GC B cells only in control mice were associated with cell activation (FDR = $6.84 \times 10^{-4}$), DNA-binding transcription factor activity (FDR = $6.84 \times 10^{-4}$) and cell cycle (FDR = $4.2 \times 10^{-3}$). By contrast, gene sets associated with mitochondria metabolism (FDR = $9.48 \times 10^{-3}$) were enriched in HuR KO GC B cells but we did not observe changes in mitochondrial mass, mitochondrial potential or ROS levels compared to control GC B cells (Supplementary Fig. 2d). Thus, our data suggest that HuR might be required for activation, proliferation and to control DNA damage in GC B cells.

**HuR controls entry and progression through the cell cycle**. Myc expression rescues GC B cell clones from apoptosis and sets a transcriptional programme that allows GC B cell transition from the LZ to the DZ as well as entry into the cell cycle[12,14]. In line with the reduced expression of Myc in positively selected HuR KO GC B cells, the global analysis of gene expression changes between LZ and DZ GC B cells showed in the absence of HuR a significant reduction in the expression of a set of genes directly regulated by Myc (Fig. 7a). Further clustering of Myc-dependent genes based on HuR binding to their mRNA transcripts revealed a significant reduction in the expression of these genes in both HuR KO LZ and DZ GC B cells compared to those genes that were not bound by HuR (Fig. 7b). This is consistent with the hypothesis that HuR controls a gene programme dependent on Myc for GC expansion.

GSEA also showed a reduced expression of genes associated with the cell cycle in HuR KO DZ B cells (Fig. 7c and Supplementary Fig. 6a, b). Detailed analysis of gene expression changes associated with each of the phases of the cell cycle showed that HuR was required for the expression of genes associated with the G1 phase and entry into S phase (Supplementary Fig. 6c). By contrast, HuR was dispensable for the expression of genes in the S, G2 and M phases as changes in the expression of HuR target genes was similar to those genes that were not bound by HuR (Supplementary Fig. 6c). *Rb1*, *Prim2*, *Ccnd2* and *Ccnd3* were among those regulators of cell cycle progression that were reduced in HuR KO GC B cells, whereas the expression of the cell cycle inhibitors at G1/S phase *Cdkn1a* and *Cdkn1b* was increased in these cells (Fig. 7d and Supplementary Fig. 7a, b). Taken together, our data suggest that HuR is required for GC B cell proliferation by directly controlling the expression of Myc and cell cycle-associated genes required for cell cycle entry and progression.

To establish the function of HuR as a regulator of GC B cell proliferation, we studied by flow cytometry the incorporation of bromodeoxyuridine (BrdU) into proliferating GC B cells from control and HuR^fl/fl AID^Cre mice immunised with NP-KLH for 7 days (Fig. 7e). The overall percentage of BrdU⁺ GC B cells in HuR^fl/fl AID^Cre mice was reduced 1.25-fold compared to control

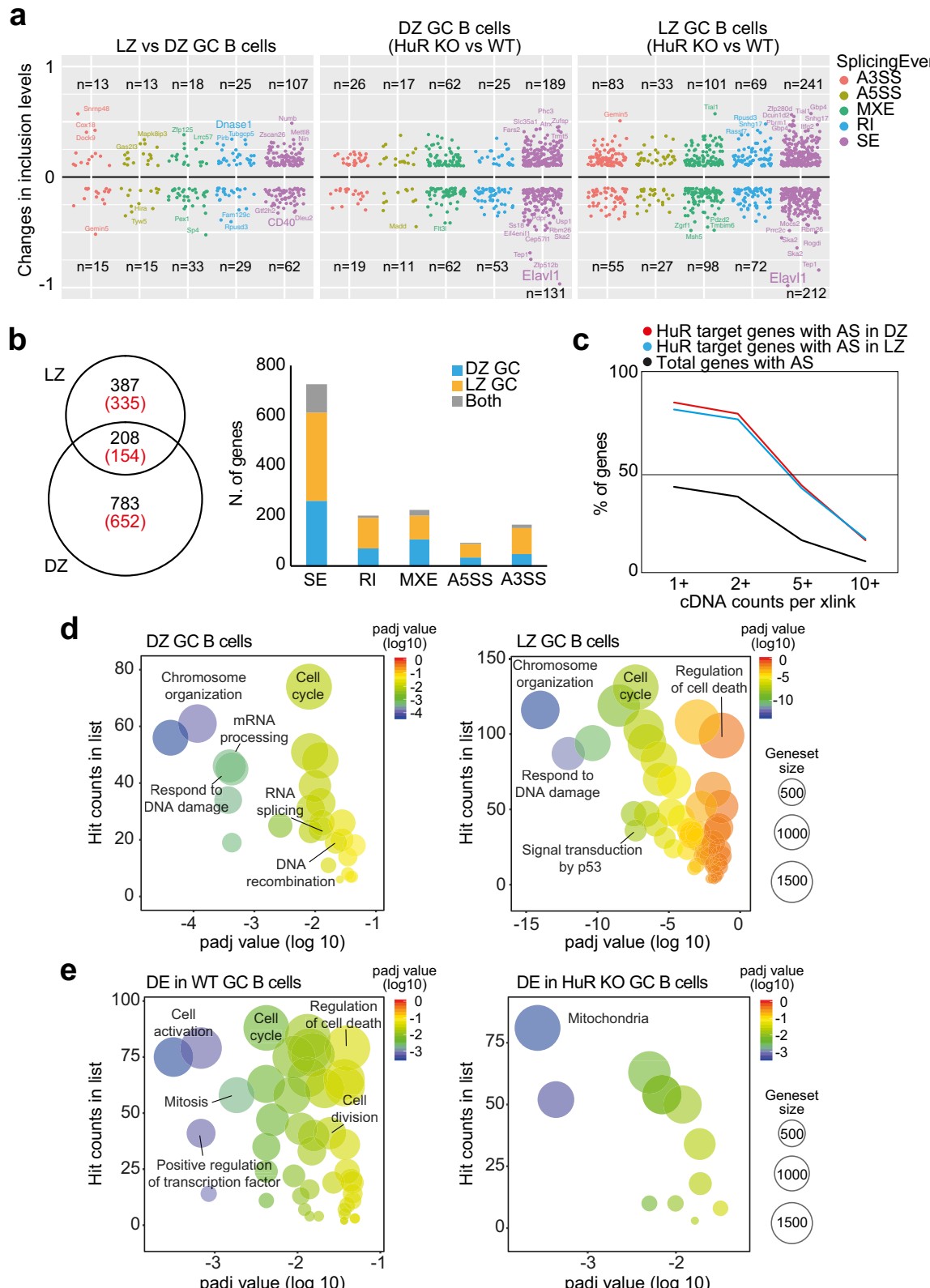

mice. This was due to a significant reduction in the percentage of proliferating DZ GC B cells but not LZ GC B cells. Sequential in vivo labelling of GC B cells with 5-ethynyl-2′-deoxyuridine (EdU) and BrdU to follow cell cycle progression showed a decrease in the percentage of HuR KO GC B cells entering S phase as well as a reduction in the transition from early to late S phase (Fig. 7f, g). Consistent with this, HuR also promoted B cell

cycle progression in vitro. Proliferation of HuR KO B cells co-cultured with 40LB stroma cells was reduced 3-fold compared to control cells (Supplementary Fig. 6d). Analysis of BrdU incorporation and DNA content showed an increased percentage of HuR KO GC B cells remaining in the G0–G1 phase with a lower proportion of cells progressing through S and G2–M phases compared to control cells (Supplementary Fig. 6e). The defect in

**Fig. 6 HuR controls mRNA splicing in GC B cells. a** Alternative mRNA splicing in DZ and LZ GC B cells in the absence of HuR. Changes in inclusion levels were calculated with rMATS and classified into skipped exon (SE), retain intron (RI), mutually exclusive exons (MXE), alternative 5′ splice site (A5SS) and alternative 3′ splice site (A3SS). Only those alternative splicing events with an inclusion change >10% and FDR < 0.05 are shown. **b** Number of alternatively splicing events (in black) and spliced genes (in red) in DZ and LZ GC HuR KO B cells. Left panel, Venn diagram showing the overlap of genes with significant splicing events in the absence of HuR. Right panel, bar plot with the total number of genes affected by alternative splicing classified by type of splicing event. **c** Percentage of genes with alternative splicing (AS) in HuR KO LZ and DZ GC B cells that are targets of HuR. The percentage of genes is classified based on the number cDNA counts mapped to unique crosslink sites (HuR iCLIP from LPS-activated B cells). **d** REVIGO plots summarising gene sets associated with differentially spliced genes in HuR KO DZ B cells (left) and HuR KO LZ GC B cells (right). **e** REVIGO plots showing gene sets enriched with the genes identified as differentially expressed only in control GC B cells or only in HuR KO GC B cells (in **d**, **e**, adjusted p values are calculated with BH method by REVIGO).

cell cycle progression in HuR KO B cells was independent of cells being cultured in the presence of interleukin (IL)-4 or IL-21 (Supplementary Fig. 6e), of the Ig isotype expressed by the cells, and of cellular differentiation into plasma cells (Supplementary Fig. 6f). In summary, HuR has a fundamental function for the expression of cell cycle genes that timely control the entry and progression of GC B cells through the cell cycle.

**HuR limits deamination in GC B cells.** GC B cell proliferation is required for Ig SHM and high-affinity antibody diversification. Our transcriptomics analysis highlighted that HuR might regulate the splicing of genes involved in DNA recombination and DNA damage repair (DDR) in GC B cells (Supplementary Data 4) without affecting mRNA abundance between HuR KO GC B cells and control cells (Fig. 8a). However, HuR KO DZ B cells failed to increase the expression of *Chek2* and other genes of the DDR associated specifically with the G1 phase of the cell cycle (Fig. 8a and Supplementary Fig. 7b). This defect could not be ascribed to a global regulation of genes selectively bound by HuR. Additionally, a gene signature associated with deamination activity was increased both in DZ GC B cells and LZ GC B cells from HuR^fl/fl AID^Cre mice compared to controls (Fig. 8b, c). Consistent with these observations, the quantitation of somatic mutations in the JH4 intron revealed an increased in the incidence and number of mutations in GC B cells from HuR^fl/fl AID^Cre mice compared to controls (Fig. 8d, e). *Aicda* mRNA expression was not increased in HuR KO GC B cells compared to control cells, but other family members such as *Apobec1* were significantly increased in the absence of HuR (Supplementary Fig. 7b).

Next, we assessed further DNA damage in HuR KO GC B cells. The expression of the DNA damage sensor *Trp53* was not detectably increased in HuR KO GC B cells compared to control GC B cells. However, the expression of *Cdkn1a* and *Cdkn1b*, two genes induced upon p53 activation in response to DNA damage, was elevated in HuR KO DZ GC B cells (Supplementary Fig. 7b). Quantitation of the DNA damage marker phospho-histone H2A.X (pS319) by flow cytometry also showed a significant increase in HuR KO GC B cells compared to control cells (Fig. 8f). This increase was significant in LZ GC B cells and class-switched antigen-specific GC B cells (Fig. 8g and Supplementary Fig. 7c). Thus, we conclude that HuR is required for optimal expression of genes involved in SHM and DDR, and in its absence, GC B cells fail to restrict DNA damage leading them into apoptosis.

## Discussion
RNA interactome capture[34] has described >1500 RBPs encoded in the mammalian cell genome[35]. However, the importance of regulation of gene expression by RBPs in the humoral response is poorly understood. This study shows the importance of the RBP HuR in initiating and maintaining GC responses and the production of high-affinity antibodies. HuR is dispensable for CSR

and the terminal differentiation of B cells, but selection, maintenance and expansion of antigen-specific GC B cells is dependent upon HuR actions in GC B cells.

Our data provide evidence that HuR is required for the full expression of Myc and Myc-associated transcriptional programmes that allow migration of LZ GC B cells to the DZ and prepares LZ GC B cells to enter into the cell cycle. The GC reaction is not sustained in the absence of HuR in a manner that resembles the collapse of GCs in the absence of Myc[11,12]. Myc protein abundance reflects the magnitude of Tfh cell help received by LZ GC B cells during positive selection by Tfh cells[14]. Activation of phosphoinositide-3 kinase (PI3K)-AKT and NF-κB signalling pathways are both required for productive upregulation of Myc in GC B cells[13]. These pathways, or BCR cell surface expression, were not found deregulated in our transcriptomics analyses or by fluorescence-activated cell sorting (FACS). PI3K or NF-κB activation has the potential to induce HuR in tumour cells[36], and they might be also responsible for the increased HuR expression in positively selected LZ GC B cells. It is likely that different transcriptional and post-transcriptional mechanisms cooperate to induce the expression of Myc and Myc-dependent genes in LZ GC B cells. We have previously shown that PTBP1 is required for mRNA biogenesis and GC B cell proliferation induced by Myc[18]. HuR associates with RNA transcripts from 134 Myc-regulated genes in B cells, in which overall mRNA abundance was increased in DZ B cells compared to LZ B cells and significantly reduced in the absence of HuR. By contrast, mRNAs not bound by HuR did not change significantly between HuR KO DZ and LZ B cells. This highlights that HuR, PTBP1 and, maybe, others[37] could be part of a more extended post-transcriptional programme that acts downstream of gene transcription for global remodelling of the transcriptome and sets the fate of GC B cells upon positive selection.

HuR might control *Myc* mRNA translation without affecting *Myc* mRNA abundance in GC B cells, although this needs further exploration. The molecular mechanism by which HuR controls *Myc* mRNA translation is cell- and context-dependent. In cell lines, HuR cooperates with microRNAs *let-7* or *miR-17-19b* to block *Myc* mRNA translation[27,28]. However, HuR promotes *Myc* mRNA translation under hypoxia[33] or in the presence of polyamines[29]. HuR is extensively regulated at the post-translational level by multiple mechanisms. Caspase-dependent cleavage[33] and phosphorylation by SphK1 and Chk2[29,38] have been suggested as potential mechanisms controlling HuR-dependent *Myc* mRNA translation and cell proliferation. Thus, it is possible that Tfh:B cell encounter not only induces a coordinated expression of Myc and HuR but also promotes HuR-dependent *Myc* mRNA translation.

Overexpression of Myc failed to rescue the GC reaction in the absence of HuR, highlighting the multiple functions of HuR in GC B cells. Our transcriptomic analysis revealed that HuR controls splicing of hundreds of mRNAs that can modulate GC responses in a Myc-independent manner. This is in line with our

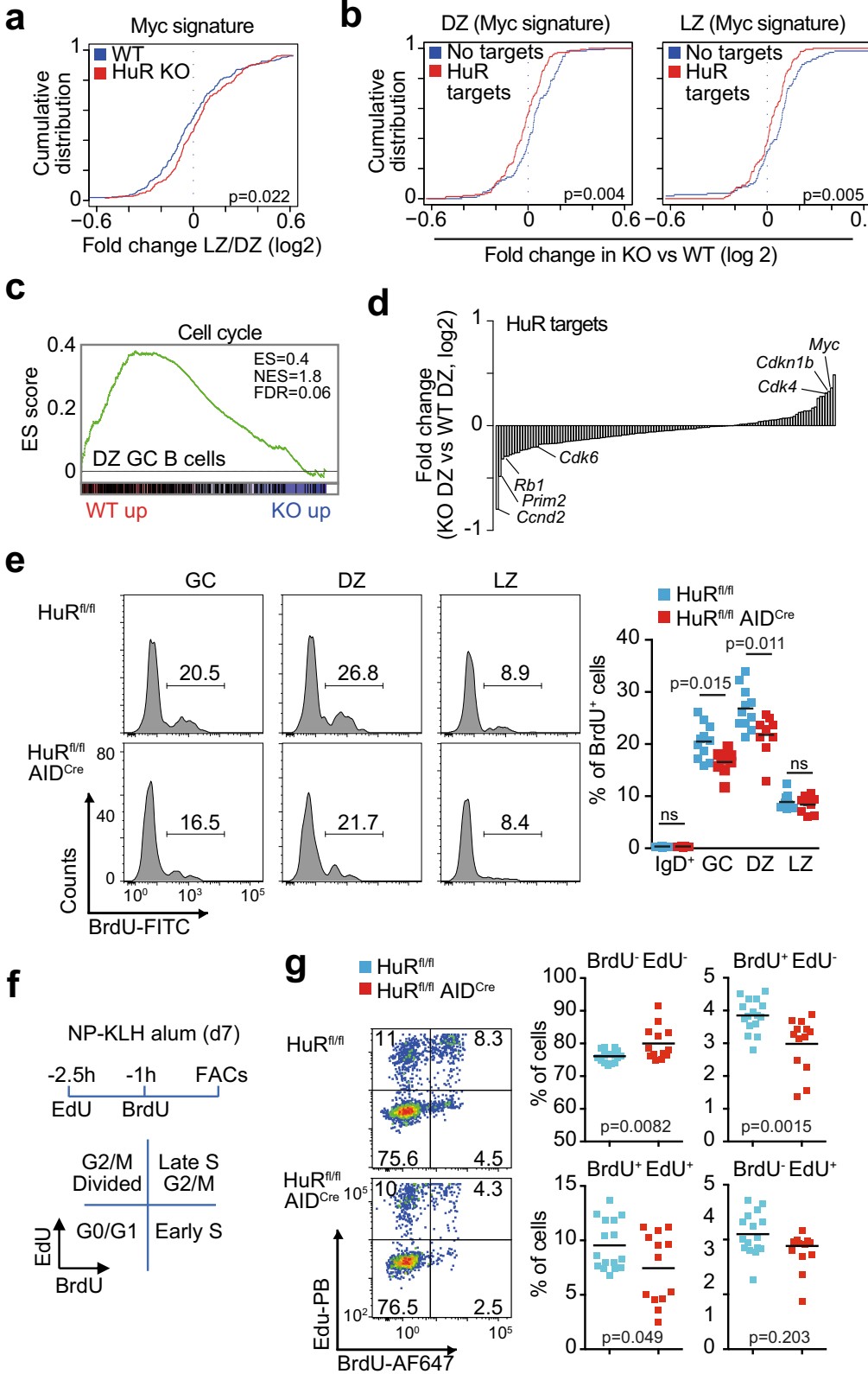

previous report showing that HuR maintains mRNA transcriptome fidelity in B cells by preventing inclusion of introns and cryptic exons[20]. HuR controls the overall abundance of its mRNA targets, although changes in mRNA expression were in general modest in HuR KO GC B cells. Changes in mRNA splicing and abundance were prevalent in RNA regulons involved in cell cycle and DNA damage responses. HuR modulates the expression of a

subset of HuR mRNA targets involved in GC B cell entry into the cell cycle and transition from G1 to S. This is of special importance as it shows that HuR not only controls *Myc* expression but can also regulate directly the fate of important genes in cell cycle, including *Ccnd2*, *Ccnd3*, *Rb1*, *Prim2* or *Ccnb1*[24,39,40]. This observation is further supported by the fact that HuR KO B cells also show a defect in proliferation and cell cycle in vitro. iGC B

**Fig. 7 HuR controls entry and progression through the cell cycle. a** Cumulative distribution of fold change in the expression of Myc-dependent genes (Hallmark gene set) in LZ and DZ GC B cells (Kolmogorov–Smirnov test). **b** Cumulative distribution of changes in the expression of Myc-dependent genes that are targets or not of HuR (Kolmogorov–Smirnov test). **c** Enrichment analysis of genes related to cell cycle (GSEA, Gene set from Reactome). **d** Quantitation from our transcriptome analysis of the changes in the expression of HuR mRNA targets associated with the cell cycle phase G1–S in HuR KO DZ GC B cells compared to control cells (only HuR mRNA targets identified in all conditions of B cells treated with LPS, αCD40+IL4 or αIgM+IL4 were considered). **e** BrdU incorporation in GC B cells from HuR[fl/fl] and HuR[fl/fl] AID[Cre] mice at day 7 after immunisation with NP-KLH in alum. Left panels, representative histograms of BrdU staining in total GC B cells or in DZ and LZ GC B cells. Right panel, percentage of BrdU+ cells in the cell populations indicated (data from 2 independent experiments, $n = 11$ (HuR[fl/fl]) or $n = 9$ (HuR[fl/fl] AID[Cre]) mice, two-sided Mann–Whitney test). **f** Strategy for labelling proliferating GC B cells in vivo with EdU and BrdU. **g** Analysis by flow cytometry of cycling GC B cells from HuR[fl/fl] and HuR[fl/fl] AID[Cre] mice at day 7 after immunisation. Left panels, representative contour plots of BrdU and EdU incorporation by cycling GC B cells. Right panels, percentage of cells labelled with EdU, BrdU and EdU/BrdU (data from 2 independent experiments, $n = 16$ (HuR[fl/fl]) or $n = 13$ (HuR[fl/fl] AID[Cre]) mice, two-sided Mann-Whitney test). Source data are provided as a Source data file.

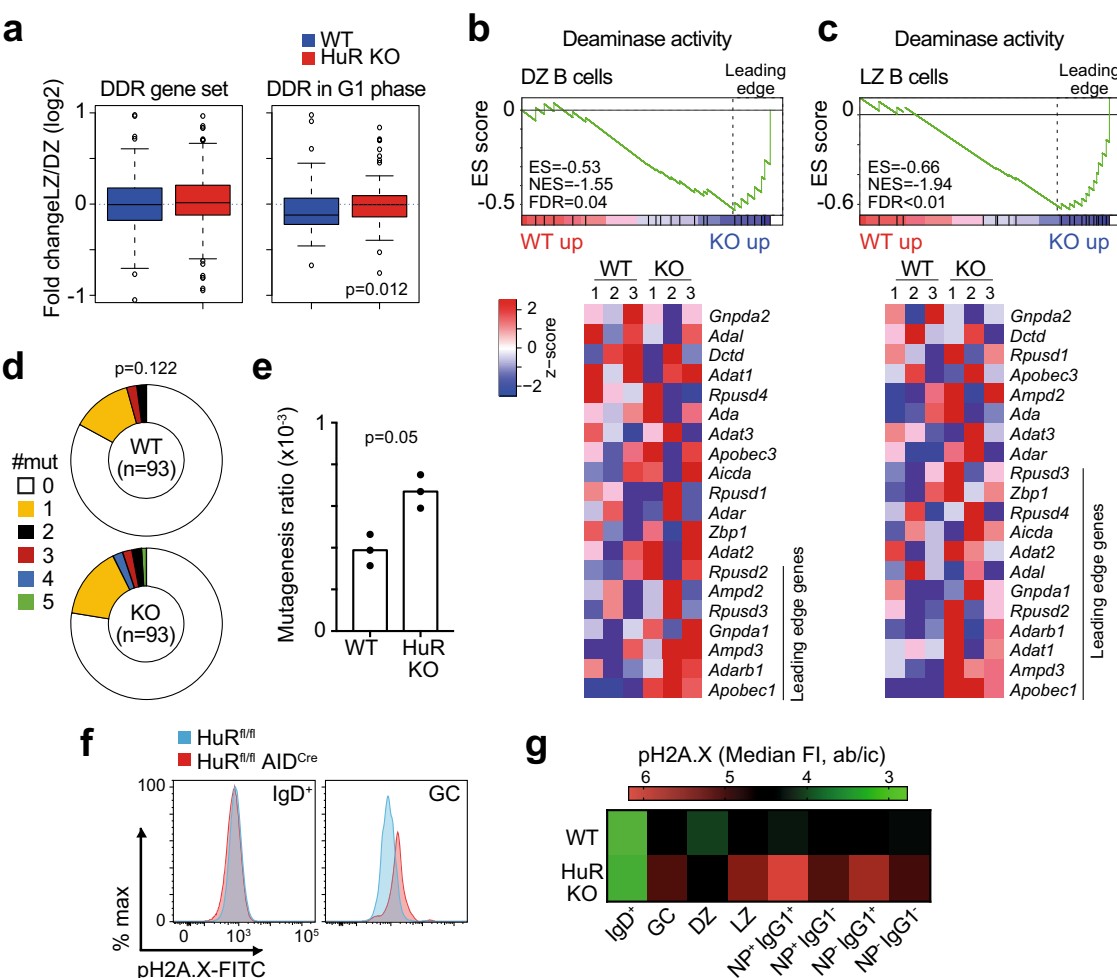

**Fig. 8 DNA damage is increased in the absence of HuR. a** Change in the global expression of genes associated with the DNA damage response ($n = 119$ genes in DDR gene set from Reactome, $n = 56$ genes in DDR in G1 phase) in LZ and DZ GC B cells (data presented as box plots showing the mean, 25–75% range and whiskers from 5 to 95%, two-sided Wilcoxon test). **b, c** Enrichment analysis of genes involved in deaminase activity (GSEA, Gene set GO:0019239) in DZ GC B cells (**b**) and LZ GC B cells (**c**). Top panels, enrichment plot showing the gene enrichment score (ES), the normalised enrichment score (NES) and the false discovery rate (FDR). Bottom panels, heatmap showing the relative expression of genes involved in deaminase activity in DZ and LZ GC B cells at day 7 after immunisation with NP-KLH in alum. Leading genes driving increased deaminase activity in HuR KO GC B cells are indicated in blue. **d** Analysis of SHM in GC B cells sorted from HuR[fl/fl] and HuR[fl/fl] AID[Cre] mice at day 7 after immunisation ($n = 93$ clones analysed per mouse genotype, two-sided Mann–Whitney test). **e** Quantitation of the mutagenesis ratio in the JH4 intron ($n = 3$ mice processed independently in different experiments, two-sided Mann–Whitney test). **f** Flow cytometric histograms showing the expression of pH2A.X (pS139) in naive and GC B cells from HuR[fl/fl] and HuR[fl/fl] AID[Cre] mice at day 14 after immunisation (data representative from 1 of the 2 independent experiments performed, $n = 6$–9 mice per group). **g** Heatmap of pH2A.X MFI in the cell types indicated (data is from 2 independent experiments, $n = 11$–16 mice per group). Source data are provided as a Source data file.

cells do not receive help from Tfh cells and do not undergo affinity maturation. Thus, they better reflect the intrinsic role of HuR in cell cycle control in the absence of Myc induction.

Numerous studies have reported the importance of HuR in cell proliferation and apoptosis[41,42]. HuR is increased in late S and G2–M compared to non-cycling GC B cells remaining at G0–G1. Moreover, HuR-dependent RNA binding, stabilisation and translation are linked to HuR subcellular location and phosphorylation. HuR shuttles from the nucleus to the cytoplasm during the cell cycle[43]. This process is controlled by the cell cycle-associated kinase Cdk1[44] and the DDR kinase sensor Chk2[29]. Induced deletion of HuR with tamoxifen increases sensitivity to gamma irradiation and cell death[42]. HuR regulates the G1/S cell cycle-regulator Cdkn1a[45] and the DNA damage checkpoint master regulator p53[46,47]. These checkpoints might be inappropriately active in GC B cells in the absence of HuR, as suggested by the increased expression of Cdkn1a and Cdkn1b, triggering GC B cell death.

AID activity and cell cycle progression are closely linked. Nuclear AID is highly stable at the G1 phase when DNA damage is mostly repaired by error-prone mechanisms[48]. Increased DNA deamination and altered DDR in the absence of HuR can have a negative effect over cell cycle progression and cell viability of GC B cells. DNA damage of Ig genes in DZ GC B cells promotes cell cycle arrest at G1–S and, if unresolved, can trigger cell apoptosis[49]. Our iCLIP data show that HuR binds to Aicda and Apobec1 mRNAs and has the potential to regulate the expression of the latter. Aicda mRNA abundance remained constant in HuR KO GC B cells and the low degree detection of Aicda exon 2 skipping was most likely due to the AID[Cre] transgene used for HuR deletion. Overactivation of AID is commonly found in GC-derived B cell lymphomas and it is associated with genome instability and increased mutagenesis[50]. The increased mutagenesis in the JH4 intron reflects hyperactivation of deaminases in HuR KO GC B cells, although the rapid loss of antigen-specific GC B cells in HuR[fl/fl] AID[Cre] mice prevented us from assessing whether increased mutagenesis contributed to affinity maturation. Similarly, the increased phosphorylation of H2A.X and apoptosis and reduced expression of Chk2 in HuR KO GC B cells suggests that HuR is a key modulator in synchronising mutagenesis, DDR and cell cycle progression. These results are in line with previous observations in which HuR is at the centre of an ATM kinase-dependent regulatory mechanism that allows DDR in B cells[51].

In summary, here we present evidence for the essential function of post-transcriptional regulation by HuR in GC maintenance and high-affinity antibody production. The interaction of GC and Tfh cells during positive selection is likely to trigger important transcriptional and post-translational changes inducing HuR expression and possibly affecting its subcellular location, binding and function. In a possible model, HuR is likely part of an extended network of RBPs that controls mRNA biogenesis, stability and translation in GC B cells. Conditional deletion of HuR in GC B cells leads to ample changes in mRNA splicing and abundance of hundreds of genes. Among these genes, we identified that HuR is required for the expression of Myc and a Myc-dependent programme that enables antigen-based GC B cell selection by Tfh cells. HuR also modulates the expression of cell cycle genes required by GC B cells to progress from the G1 to the S phase of the cell cycle. It is then possible that the extended permanence of cells into the G1 phase leads to an increased residence of AID in the nucleus and mutagenesis[52]. This, in combination with high expression of Apobec1 in HuR KO GC B cells, will result in increased DNA damage and cell death and the loss of antigen-specific GC cells that differentiate into plasma cells producing high-affinity antibodies. Deeper study of how HuR

controls the expression of important transcriptional and post-transcriptional programmes in the GC reaction is certainly needed and it will open new opportunities to understand further the complex molecular biology of humoral immune responses.

## Methods

**Mice and animal procedures**. Elavl1[tm1dkon] (HuR[fl/fl]) mice[53] were crossed with Tg(Aicda-cre)9Mbu (AID[Cre]) mice[54] or with CD79a[tm1(cre)Reth] (Mb1[Cre]) mice[55]. HuR[fl/fl] AID[Cre] mice were crossed with Myc[tm1Slek] (Myc[GFP/GFP])[31] or with Gt(ROSA)26Sor[tm13(CAG-MYC,-CD2*)Rsky 11]. B6.Rag2[tm1.1Cgn] (Rag2-KO) and C57BL/6-Ly5.1 (CD45.1[+]) mice were kindly provided by Dr. A. Saoudi and Dr. N. Blanchard (both from INFINITy, Toulouse, France). All mice were maintained on a C57BL/6 background strain. The number of animals was decided on the basis of preliminary data and statistical power calculations. Data fulfil a log distribution, the null hypothesis is that the medians of the two samples are identical and assuming a 2-fold difference between means, a standard deviation of 1/3 of the mean value, a ratio between groups of 1 and a type I error rate of 5% we need a minimum of 6–7 mice per group and experiment for a 90% statistical power. Mice of 8–16 weeks of age were used. Littermate control mice (Cre negative) were used in most experiments. No animals were excluded due to a lack of responsiveness to immunisation. Randomisation, but not experimental 'blinding', was set in these studies. All animal procedures at the Babraham Institute were approved by the local animal welfare and experimentation committee and by the UK Home Office. Experiments at INFINITy were approved by the local ethical committee and by the French Ministry of Education, Research and Innovation (project number 16731-2018090610469877v4). No primary pathogens or additional agents listed in the FELASA recommendations have been confirmed during health monitoring surveys of the mouse stock holding rooms maintained in the Babraham Institute Biological Support Unit and at the CREFRE in Toulouse since the opening of these barrier facilities. Ambient temperature was ~19–21 °C and relative humidity 52%. Lighting was provided on a 12-h light: 12-h dark cycle, including 15 min 'dawn' and 'dusk' periods of subdued lighting. After weaning, mice were transferred to individually ventilated cages with 1–5 mice per cage. Mice were fed CRM (P) VP diet (Special Diet Services) ad libitum and received seeds (e.g. sunflower, millet) at the time of cage cleaning as part of their environmental enrichment.

Immune responses were analysed upon intraperitoneal (ip) or subcutaneous (sc, in the hook) injection of 100 or 20 µg of NP-KLH (Biosearch Technologies) precipitated in aluminium hydroxide gel (Serva). No adjuvant was used to analyse NP-KLH recall responses. Alternatively, immune responses were elicited using sheep red blood cells (SRBCs; 2 × 10[8] SRBCs in Alsevers, ip, TCS Bioscience). For in vivo analysis of cell cycle, animals were injected (ip) with 0.5 mg of EdU (Sigma Aldrich) and 1.5 h later with BrdU (2 mg, Sigma Aldrich). Animals were analysed 2.5 h after first injection. Alternatively, mice were injected with BrdU only and culled 2.5 h later.

BM chimeras were performed as previously described[18]. Briefly, BM cells were isolated from tibias and femurs of C57BL/6-Ly5.1 (CD45.1[+]) mice and HuR[fl/fl] or HuR[fl/fl] AID[Cre] (both CD45.2[+]) mice, treated with ACK lysis buffer (Thermo Scientific) to remove red blood cells and mixed in a 1:3 ratio (CD45.1/CD45.2). In all, 5 × 10[6] cells were intravenously injected into Rag2-KO mice previously irradiated sublethally (X-ray irradiation, 500 Gy). Experimental analyses were performed after reconstitution 8 weeks later.

**Enzyme-linked immunosorbent assay (ELISA) and enzyme-linked immuno-spot assay (ELISPOT)**. NP-specific antibodies were detected by ELISA as previously described[56]. NP-specific antibody endpoint titres were used as a measure of relative concentration. NP-specific ASCs were detected by ELISPOT as described[57]. Antibodies used are described in Supplementary Table 1.

**B cell isolation and cell culture**. B cells from spleen or peripheral LNs were isolated using the B Cell Isolation Kit from Miltenyi Biotec (Cat. No. 130-090-862). B cells were cultured at a density of 0.5 × 10[6] cells per ml in RPMI-1640 medium (Dutch Modification) plus 10% foetal calf serum, antibiotics, 2 mM L-glutamine, 1 mM sodium pyruvate and β-mercaptoethanol (50 µM). Mitogen activation of B cells was achieved with LPS (10 µg/ml, Escherichia coli strain 0127/B8, Sigma Aldrich), anti-CD40 (10 µg/ml, clone 3/23; purified in-house) and/or anti-IgM (F(ab)2 fragment; 5 µg/ml, Jackson ImmunoResearch) in the presence of recombinant mouse IL-4 or hBaff (both from Peprotech).

In vitro derived GC B cells were generated as previously described[30]. Briefly, 17.5 × 10[3] cells were co-cultured over a monolayer of 40LB stroma cells irradiated (120 Gy) and seeded in a 24-well plate 24 h before. IL-4 (2 ng/ml) was added to the RPMI medium that was replaced every 2 days. At day 4 of cell culture, iGC B cells were recovered and seeded on freshly irradiated 40LB stroma cells. At day 4, IL-4 (2 ng/ml) or IL-21 (10 ng/ml, Peprotech) was added to the media. Analysis of cell cycle was performed after adding 10 µM BrdU 30 min before harvest.

**Flow cytometry and cell sorting**. Analysis of B cell populations was performed using specific antibodies as indicated in Supplementary Table 1. GC B cell subsets were identified as follows: GC B cells—CD19[+] IgD[−] CD38[−] CD95[+], DZ GC B cells—

CD19+ IgD− CD38− CD95+ CXCR4+ CD86−. LZ GC B cells—CD19+ IgD− CD38− CD95+ CXCR4− CD86+. Myc+ GC B cells were identified using antibodies against c-Myc and GFP. CSR and HuR expression were analysed using specific antibodies against IgM, IgG1 and HuR. NP (LGC Biosearch Technologies) directly conjugated to PE.

For assessment of cell viability, we incubated LN cells with CaspACE™ FITC-VAD-FMK in Situ Marker (1 μM per 10^7 cells) for 30 min at 37 °C in complete RPMI medium. Zombie NIR Fixable Viability was used to stain dead cells and in combination with an anti-mouse Fc Receptor Blocking antibody (clone 2.4G2) for 15 min at 4 °C prior to staining of cell surface markers. For intracellular staining, cells were fixed and permeabilised with the BD Cytofix/Cytoperm™ Fixation and Permeabilization Solution from BD Biosciences. EdU staining was carried out after DNA digestion with TurboDNase (ThermoFisher Scientific) using the Click-iT™ EdU Pacific Blue™ Flow Cytometry Assay Kit (Cat. No. C10418, ThermoFisher Scientific) followed by incubation with Anti-BrdU antibody [clone MoBu-1]. BrdU and DNA staining were carried out using the FITC BrdU Flow Kit from BD Biosciences (Cat. No. 559619). Data were collected using a BD Fortessa and analysed using the software FlowJo vX.

**Imaging.** Spleens from immunised mice was embedded in Tissue-Tek O.C.T. (Sakura Finetek) before snap-freezing and cryopreservation. Ten-micrometre sections were obtained in a Cryostat, fixed with acetone for 15 min at −20 °C, dried at room temperature, blocked with phosphate-buffered saline (PBS) with 0.3% Triton X-100 and 5% bovine serum albumin for 1 h. at room temperature. Staining with specific antibodies against IgD (coupled to AF488, 1 in 400), Ki67 (coupled to AF488, 1 in 100) and CD21/35 (coupled to biotin, 1 in 400) was performed overnight at 4 °C in a wet chamber. After washing, tissue samples were incubated with streptavidin-PE (1 in 800) for 45 min at room temperature. Sections were then washed (3×) with PBS with 0.3% Triton X100, dried and mounted on slides using ProLong Gold Antifade Mountant (Thermo Scientific). Images were collected in a Leica SP8 confocal microscope using the Leica Application Suite X (LAS X) software and analysed with the FIJI Image J software.

**Analysis of SHM.** Mutations in the JH4 intronic region were analysed as previously described[18]. Briefly, DNA was extracted from FACS-sorted GC B cells at day 7 after mouse immunisation with NP-KLH alum. GC B cells from individual mice (n = 2 per group) were analysed. JH4 intronic regions were amplified by PCR using PfuUltra II Fusion HS DNA Polymerase (Agilent Technologies), primers JH4 (fwd), TCCTAGGAACCAACTTAAGAGT and JH4 (rev) TGGAGTTTTCT-GAGCATTGCAG (Supplementary Table 2) and 35 cycles of denaturation 95 °C (30 s), annealing 57 °C (30 s) and extension 72 °C (30 s). cDNA was cloned using the Zero Blunt® TOPO® PCR Cloning Kit (Cat. No. 450245, ThermoFisher Scientific). Mutation frequencies were calculated by dividing the total number of mutations identified in each replicate by the total length of amplified DNA (amplicon of 565 bp).

**RNA sequencing.** Transcriptomics analysis of LZ and DZ GC B cells from HuR^fl/fl and HuR^fl/fl AID^Cre mice immunised with NP-KLH alum for 7 days was performed by FACS cells as previously described[18] and summarised in Supplementary Fig. 8. Briefly, GC B cells were pre-enriched by depleting IgD+, CD3e+, Gr1+ and Ter119+ cells with biotinylated antibodies, anti-biotin magnetic beads and MACS LS columns. GC B cells were labelled with specific antibodies (LZ GC B cells = CD19+ CD38− CD95+ GL7+ CXCR4− CD86+; DZ GC B cells = CD19+ CD38− CD95+ GL7+ CXCR4+ CD86−). Four biological samples (two from males and two from females) for each genotype were sorted and RNA was isolated using the RNeasy Micro Kit from Qiagen (Cat. No. 74004). RNA quality was analysed on a 2100 Bioanalyzer (Agilent) and libraries were prepared with 50 ng of RNA using the SureSelect Strand-Specific RNA Library Preparation Kit for Illumina (Cat. No. G9691A, Agilent). Libraries were multiplexed and sequenced across three lanes on an Illumina HiSeq 2500 platform (100 bp paired-end mode). Other RNA sequencing data sets used in this study are previously described (GSE109732, GSE82003, GSE80669)[7,32,58].

**Bioinformatics.** Sequencing reads were trimmed using TrimGalore (v0.4.4) and aligned to the mouse genome (GRCm38) with HiSat (v2.1.0). Alignment rate was >93%. Bam files were merged using SamTools (v1.8) and counted using Feature-Counts (v1.6.3), Mus_musculus.GRCm38.89.gtf annotation and default parameters. R (v3.4.1) and DESeq2 (v1.24.0) was used for differential expression analysis. Conditions included genotype, sex of animals and sample preparation day to control for variation in the data due to these parameters. Changes in gene expression with a FDR < 0.05 was considered as significant. Differential AS was analysed with rMATS (v4.0.2)[59]. Sequencing data were previously trimmed with Flexbar (v3.0) and SamTools (v1.8) and remapped using STAR (v 2.5.1b), an splicing aware tool (Python (v2.7.2)). Read length was 101, parameter -paired and annotation Mus_musculus.GRCm38.90.gtf. Differentially spliced events in pairwise comparison were considered significant if FDR < 0.05 and have an absolute inclusion level of >0.1.

In vivo interactions of HuR and RNA in splenic B cells treated with LPS for 48 h has been previously described (GSE62148)[20]. In parallel, naive B cells were treated

or not with αCD40+IL4 or αIgM+IL4 to detect HuR physical interactions with Myc 3'UTR. Briefly HuR:RNA interactions were detected using iCLIP and sequencing. After annotation of reads to the mouse genome, nucleotide −1 was annotated as the crosslink binding site. Peak call analysis was performed as described[60]. The number cDNA counts associated with a single crosslink site was quantified based on read length and the sequence of the 5' barcode. Annotation of crosslink sites into different genomic features (introns, exon, 5'UTR, 3'UTR or CDS) was performed with R (v3.4.1), bioMart (v2.40.4), GenomicRanges (v1.36.0) and Mus_musculus.GRCm38.90.gtf. Only those crosslinks with a FDR < 0.05 and >5 cDNA counts annotated to the same nucleotide were considered as significant (if not stated otherwise).

Gene ontology enrichment analysis was performed using either ToppGene[61] or GSEA,[62] using default settings. Selected gene sets were from Hallmark, Reactome or AmiGO2. Data from our sequencing data sets and gene sets were extracted and plotted in R (v3.4.1) using ggplot2 (v3.2.1).

**Statistics and reproducibility.** Statistics were performed in R or using Prism. Statistical tests performed are indicated in each figure legend. Briefly, Mann–Whitney or Wilcoxon tests were used for pairwise comparisons; analysis of variance was used when having more than two groups to compare. Two-sided analyses were performed in all cases. Kolmogorov–Smirnov test was used to assess equality of one-dimensional probability cumulative distributions. Benjamini and Hochberg test was used for multiple testing and FDR calculation (if not stated otherwise). All experiments were reproduced at least twice successfully and whenever possible.

**Reporting summary.** Further information on research design is available in the Nature Research Reporting Summary linked to this article.

## Data availability

The RNAseq data that support the findings in this study have been deposited in Gene Omnibus (GEO) with the primary accession code GSE145413. Public source RNAseq and iCLIP data sets analysed in this study were obtained from GEO with the accession numbers GSE109732, GSE80669, GSE82003 and GSE62148. Any other data or material used in this study are available from the corresponding author upon reasonable request. Source data are provided with this paper.

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

## Acknowledgements
We thank D. Kontoyiannis for the HuR^fl/fl mouse, M. Reth for the Mb1^Cre mouse, M. Busslinger for the AID^Cre mouse and K. Rajewsky for the ROSA26-Myc mouse. We thank K. Tabbada, from Babraham Institute next generation sequencing facility, and K. Bates, D. Sanger and all personnel from the Biological Services Unit and flow cytometry facilities of The Babraham Institute, Toulouse institute for infectious and inflammatory diseases (INFINITy) and GenoToul for technical assistance. We also thank Sophie Allart and Simon Lachambre for their help at the imaging core facility in INFINITy. This study was supported by funding from the Biotechnology and Biological Sciences Research Council (BBSRC) (BB/J00152X/1; BBS/E/B/000C0407; and BBS/E/B/000C0427 and the BBSRC Core Capability Grant to the Babraham Institute), a Wellcome Investigator (award 200823/Z/16/Z) to M.T. M.D.D.-M. is supported by funding from ATIP-Avenir-Plan Cancer (C18003BS), ANR (ANR-20-CE15-0007), foundation ARSEP R19201BB, the foundation ARC, La Ligue contre le cancer, the INSPIRE project (funded by Feder/LaRegion Occitanie, Inserm and CHU Toulouse) and Inserm-CNRS-University Paul Sabatier. D.C.-S. is supported by Boehringer Ingelheim Fonds.

## Author contributions
M.D.D.-M., I.C.O.-G., D.C.-S. and S.E.B. designed, performed and analysed experiments. M.D.D.-M. and M.L. performed bioinformatics analyses. M.D.D.-M., I.C.O.-G., S.E.B., D.L.K. and M.T. interpreted the data. M.D.D.-M. and M.T. conceived, supervised, administrated and raised funding for the project. M.D.D.-M. wrote the manuscript with input from all the authors.

## Competing interests
The authors declare no competing interests.
