## [Peer Review File · Nature Communications]

The RNA-binding protein HuR is required for maintenance of the germinal centre responseREVIEWER COMMENTS

Reviewer #1 (Remarks to the Author):

This is a lengthy and well presented study that is challenging to read. But the subject is likely to be of broad interest. I found the data as presented to be of high quality with thorough statistical precision. My minor concern is that some references could be adjusted to include presidency of my own work (Keene): 1) the last reference to Wang et al. EMBOJ 2000 demonstrated a role for HuR in cell cycle, and a paper by Tenenbaum et al., PNAS 2000 that should also be cited, was the first demonstration of multiple HuR mRNA targets including Myc and other mRNA targets of the cell cycle. In addition, a paper by Mukherjee et al. Mol Cell 2011 was published back-to-back and is usually co-cited with the reference of Lebedeva et al. 2011. I leave these to the authors' discretion.

Reviewer #2 (Remarks to the Author):

In their paper, Osma-Garcia et al investigate the role of the RNA-binding protein HuR on the GC B cell response. Regulation of RNA splicing and stability is emerging more and more as a key regulator mechanism for immune cell differentiation and survival but is still not well understood. The authors show that HuR is essential for the expansion and maintenance of the GC response and the formation of high-affinity IgG antibodies to immunisations. They link this to a loss of DZ GC B cells due to increased apoptosis. Furthermore, they show that help from Tfh cells drives expression of Elavl1/HuR. Further extensive RNAseq and iCLIP experiments are then presented to uncover the underlying mechanism.

Figure 5 collectively shows that LZ and DZ GC b cells from HuR KO mice have a differential expression of many genes but given that there is no enrichment of differentially expressed genes in genes previously identified as being bound by HuR this seems to be a fully secondary effect to the defect in GC expansion/maintenance caused by the loss of HuR but not the cause of it. This contrasts with data shown in Figure 6 where HuR-bound gene show a modest 2-fold enrichment for alternative splicing events.

Through these experiments the authors try to link the observed defects to a failure of Myc-expression in the absence of HuR. However I find this link the weakest point of the paper particularly because the forced expression of Myc does not seem to rescue the loss of GC B cells. This suggests that a failure to maintain Myc expression is at best required for some of the observed effects but clearly not sufficient. And in the absence of demonstrating that forced expression of Myc rescues some specific cellular defects, for example the progression through the cell cycle, the effect on Myc expression seems to largely irrelevant for the loss of GC B cells in the absence of HuR.

The increased mutation rate and DNA damage in the absence of HuR is interesting and provides an alternative explanation for the loss of GC B cells. It would be interesting to see if the increased number of mutations contribute to affinity maturation.

Overall this is a well-performed study that provides many intriguing observations but I am not convinced that the underlying mechanism has been found.

Specific points:

The data for a direct role of HuR in regulating Myc expression in GC B cells is quite weak. For data shown in Figure 4E, pre-gates would be helpful. The calculation of MFI on cells pre-gated on the same marker is also not appropriate because this will lead to easily skewed data based on small numbers of outliers. For the MFI plot, the authors should show both of their datasets and use an appropriate test (data doesn't seem to be normally distributed) that takes into account their full dataset from both experiments.

For dataset that combine data from multiple experiments where the experiment may introduce a shift in the overall level of the response (for example when showing both datasets in Figure 4E and possibly in Figure 7G) I would suggest the use of a linear mixed models with experiments as a blocking factor.

The authors claim that HuR is dispensable for CSR and terminal diff into PC (lines 156) based on in vitro data. However, data in Figure 1C and 3A shows a much more pronounced effect on IgG1 ASCs (or GC B cells) than IgM+ cells. Given that CSR is happening independent of the GC (Roco et al, Immunity 2019) it seems likely that both processes are affected, at least for T-dependent responses.

A key experimental mouse model for their paper is the use of AID-Cre mice. And while these mice have been widely used to delete genes selectively in GCs, the difference in cell number between mice using AID-Cre compared to Mb1-Cre (Supplementary Figure 1D) seems to indicate that the very early phase of the GC response (when HuR will be absent in Mb-1-cre mice but not necessarily in AID-Cre mice) seems to indicate a more fundamental role already in the very early establishment phase of the GC response.

In Figure 1C the authors need to show the CD45.1/C45.2 ratio in non-immunised control mice from the same experiment or the ratio in other cells within the same animal, for example T cells or non-GC B cells to demonstrate that the mice were reconstituted with a 1:3 ratio as stated. Statistical comparisons (as done in Figure 3D) should only be made between cell populations within the same animal to exclude unequal reconstitution effects between groups of mice receiving different donor BM mixes.

What is the evidence for the claim (line 191) "HuR was efficiently deleted in over 90% of GC B cells"? Data in Suppl Figure 1b seems to show around 25% HuR-+ GC B cells?

Minor editorial points:

Number of NP-specific ASCs only shown for spleen, not BM (Fig 1C, line 134)

Can the authors please clarify the immunisation schedule? In some parts they write "immunised with NP-KLH alum for 7 days". I assume this means analysis 7 days after the immunisation and not daily immunisation for 7 days?

Line 371: the 1.5-fold reduction seems very generously rounded. Based on the data in the Figure a 1.25x reduction seems more appropriate.

Line 375: Data seems to refer to Figure 7F and G.

Reviewer #3 (Remarks to the Author):

The manuscript "The RNA-binding protein HuR is required for maintenance of the germinal centre response" performs conditional deletion of the HuR to address the role of this RNA-binding protein in germinal centres. The study shows that HuR deficiency severely impairs the germinal centre reaction at different levels and through multiple mechanisms. The same authors had previously shown that HuR depletion at an earlier stage of B cell differentiation had a great impact on splicing, which led to defective mitochondrial metabolism and promoted B cell death. This new work provides a detailed and convincing characterization of the role of HuR in germinal centres. The results are novel and very interesting and the experimental approaches are appropriate. This is a solid piece of work.

Comments and suggestions

1. One general comment on the manuscript regards the relative complexity of the phenotype observed and the molecular mechanisms proposed. The study unfolds the regulatory role of HuR on *myc*, but also suggests the direct regulation of *myc*-regulated genes by HuR. In addition, HuR regulates the abundance and splicing of numerous RNAs. The combination of these molecular events promotes, on one hand, a defect in the generation of high affinity antibodies (maybe in relation with *myc* and T cell help), and on the other, proliferation and possibly survival defects. A unifying model including these concepts would be very useful. Likewise, the discussion could expand a bit more on the interconnection between these different molecular mechanisms in the

context of the germinal centre B cell (say, versus T-cell independent outcomes).

2. The study uses HuRfl/fl and HuRfl/fl AIDcre as control and HuR deficient groups of mice, respectively. Ideally, both groups should carry the Cre allele to exclude rare lateral effects of Cre expression, thus in this case, the optimal setting would have been to compare HuR+/+ AIDCre versus HuRfl/fl AIDcre mice. To dispel an effect of the Cre (and the AID transgene), it would be important to show some of the key findings (for instance, reduction of germinal centre size and high affinity antibodies) using HuR+/+ AIDCre mice as controls.

3. Figure 2 very nicely shows the germinal centre defect in HuR deficient mice. These results already suggest that proliferation (presumably in the DZ) and survival are defective. This could be directly addressed (here or later, when the cell cycle phenotype is further analyzed, at figure 7) with classical in vitro stimulation experiments (LPS, anti-CD40 IL4, etc). That would provide a more precise measure of cell division (violet cell trace or CFSE) in combination with an analysis of isotype switch. The latter is also important in the context of the deamination signature and mutations shown in Figure 8. Also, the authors may want to more precisely measure cell death (caspase, tunnel).

4. In figure 5E, it would appear that no-HuR targets tend to be more highly expressed in KO DZ than in WT DZ (just the opposite as for HuR targets). If that is how the result is read, what is the interpretation?

5. Figure 5 and 6 put show the effect of HuR defect in RNA abundance and in alternative splicing. Figure 6D and E connect these two mechanisms by showing common cellular pathways involved in both. However, it would be nice to have more information or discussion on the mechanistic connection as well. What is the overlap between these groups of genes? How can the effect of altering alternative splicing be predicted: can it lead (in some cases) to mRNA stability and thus lower abundance?

6. AS genes (Table S3) include *Aicda*, whose abundance is however not changed (Figure S7). There is a slight concern that transcripts arising from AIDCre transgenic construct could influence this AS finding. Could this be clarified?

7. Experiments on Figure S6 D-F are performed using mb1-Cre driver. Is there any reason for this? Classical in vitro experiments, ideally using HuR+/+ AIDCre versus HuRfl/fl AIDcre mice would more precisely inform on cell division (please see points 2 and 3 above).

8. The results in figure 8 are somehow difficult to interpret. Figure 8C shows a deaminase signature, but the clustering is not obvious: there seems to be disparity among the 3 replicates of each samples, and there is not a clear pattern across different samples. Can this be supported with statistical or PCA analysis? Given that *Aicda* mRNA levels are not altered, how is the signature (ie other deaminases) relevant to the somatic hypermutation phenotype? Conversely, how can the increased somatic hypermutation be explained? Is this also accompanied by increased switch recombination? (please see point 3 above) Are the levels of AID protein increased?

Minor comments

1. Nomenclature of mouse strains used in the study is at times confusing. This is particularly so in Figure 4C. Does the top panel show HuR expression in immunized MycGFP/GFP? Figure legend is read somehow differently. Is the lower graph a quantification of the representative plots shown in the upper panel?

2. I may have missed this information: for the interpretation of the data in Figure S6B and C, it would be very useful to include as supplementary information the list of genes included in the G1 to S, S and G2 to M categories, as well as their expression in LZ and DZ cells.

3. Figure 6B. Please explain in figure legend the figures shown in the Venn diagram; possibly black refers to AS events and red refers to genes?

4. Please show statistics for mutation analysis shown in Figure 8D-E.

Point-by-point response to reviewers

We thank all reviewers for their insightful comments and hope that our responses have improved the manuscript so that it is now worthy of publication.

Reviewer #1 (Remarks to the Author):

This is a lengthy and well-presented study that is challenging to read. But the subject is likely to be of broad interest. I found the data as presented to be of high quality with thorough statistical precision. My minor concern is that some references could be adjusted to include presidency of my own work (Keene): 1) the last reference to Wang et al. EMBOJ 2000 demonstrated a role for HuR in cell cycle, and a paper by Tenenbaum et al., PNAS 2000 that should also be cited, was the first demonstration of multiple HuR mRNA targets including Myc and other mRNA targets of the cell cycle. In addition, a paper by Mukherjee et al. Mol Cell 2011 was published back-to-back and is usually co-cited with the reference of Lebedeva et al. 2011. I leave these to the authors' discretion.

Response – We thank the reviewer for the positive reception of our manuscript. We have revised the manuscript to cite the literature suggested (lanes 86, 90, 474).

Reviewer #2 (Remarks to the Author):

In their paper, Osma-Garcia et al investigate the role of the RNA-binding protein HuR on the GC B cell response. Regulation of RNA splicing and stability is emerging more and more as a key regulator mechanism for immune cell differentiation and survival but is still not well understood.

The authors show that HuR is essential for the expansion and maintenance of the GC response and the formation of high-affinity IgG antibodies to immunisations. They link this to a loss of DZ GC B cells due to increased apoptosis. Furthermore, they show that help from Tfh cells drives expression of Elavl1/HuR. Further extensive RNAseq and iCLIP experiments are then presented to uncover the underlying mechanism.

Figure 5 collectively shows that LZ and DZ GC B cells from HuR KO mice have a differential expression of many genes but given that there is no enrichment of differentially expressed genes in genes previously identified as being bound by HuR this seems to be a fully secondary effect to the defect in GC expansion/maintenance caused by the loss of HuR but not the cause of it. This contrasts with data shown in Figure 6 where HuR-bound gene show a modest 2-fold enrichment for alternative splicing events.

Response – We thank the reviewer for the comments and suggestions. This insightful review has helped us to improve the manuscript and better communicate our findings.

Expression of HuR in GC B cells shapes the cell transcriptome both quantitatively and qualitatively. HuR is a multi-functional RNA binding protein that binds to thousands of RNAs and it is associated with the “fine-tuning” expression of functionally-related genes. Indeed, our studies in B cells (Diaz-Munoz et al. Nat Immunol. 2015 Apr;16(4):415-25. and this report) suggest that HuR modestly modulates the mRNA abundance of a selected number of mRNA targets but it has a significant impact on the overall expression of HuR-bound mRNAs in GC B cells, especially of those associated with the Myc and cell cycle gene signatures (as reported in the current manuscript).

Deregulation of the Myc- associated gene signature has a profound impact in the GC B cell transcriptome, making difficult to ascribe direct effects to the loss of HuR. However, global analysis of the abundance of mRNAs targeted in the 3'UTR by HuR compared to non-targeted mRNAs reveals that HuR regulates the abundance of these mRNAs (Fig 5D and 5E; Suppl. Fig 5E and 5F). In addition, HuR acts as an important splicing regulator in GC B cells, with hundreds of genes undergoing aberrant splicing in the absence of HuR. Changes in mRNA splicing and abundance were prevalent in RNA regulons involved in cell cycle and DNA damage responses, two essential cell pathways for GC responses that are found deregulated in GC B cells in the absence of HuR.

Through these experiments the authors try to link the observed defects to a failure of Myc-expression in the absence of HuR. However, I find this link the weakest point of the paper particularly because the forced expression of Myc does not seem to rescue the loss of GC B cells. This suggests that a failure to maintain Myc expression is at best required for some of the observed effects but clearly not sufficient. And in the absence of demonstrating that forced expression of Myc rescues some specific cellular defects, for example the progression through the cell cycle, the effect on Myc expression seems to largely irrelevant for the loss of GC B cells in the absence of HuR.

Response – Our data suggest that HuR is required for Myc expression (Fig. 4E and new Fig. 4F). In addition to this, HuR directly contributes to the expression of Myc-associated genes (Fig. 7B). HuR associates with RNA

transcripts from 134 Myc-regulated genes. This comprises more than 50% of the 264 genes annotated in the Hallmark Myc-associated gene signature. The overall mRNA abundance of these 134 genes is increased in DZ B cells compared to LZ B cells and significantly reduced in the absence of HuR. mRNAs that are not bound by HuR do not change significantly between DZ and LZ B cells. Taken together, this highlights the idea that HuR is part of an extended high-grade regulatory network that controls gene expression at the post-transcriptional level. Consequently, the failure to rescue GC responses in HuR conditional KO mice by enforcing the expression of the transcription factor Myc suggests that this post-transcriptional network acts, as expected, downstream of Myc-dependent transcription. In conclusion, our observations suggest for the establishment of Myc-regulated gene expression programs, the control of mRNA splicing and abundance by HuR is critical. This speaks to the emerging concept of Myc coordinating gene expression networks at all levels and is consistent with the very recent work of others (Ciesla et al. Mol Cell. 2021 Apr 1;81(7):1453-1468). (Discussion has been modified to reflect these concepts, lanes 444-450)

The increased mutation rate and DNA damage in the absence of HuR is interesting and provides an alternative explanation for the loss of GC B cells. It would be interesting to see if the increased number of mutations contribute to affinity maturation.

Response – We agree on the fact that increased mutagenesis and altered DNA damage repair might contribute to the loss of GC B cells in HuR cKO mice. However, we show multiple lines of evidence indicating that selection of antigen-specific GC B cells and production of high-affinity antibody clones is severely impaired in these mice. The failure to maintain GC responses in the absence of HuR impedes us to test whether increased mutagenesis contribute to affinity maturation. This is now indicated in the manuscript discussion, lanes 504-506 .

Overall this is a well-performed study that provides many intriguing observations but I am not convinced that the underlying mechanism has been found.

Response – Post-transcriptional control of gene expression is an emerging field and the relevance for immunity is starting to be appreciated. Here we provided evidence that HuR controls mRNA abundance and splicing of hundreds of genes in GC B cells. As such, this is the mechanism. In this manuscript we have gone further and elucidated the networks of genes regulated by this mechanism.

HuR is required for the full expression of Myc- and cell cycle- associated programs necessary for GC maintenance and affinity maturation. In this study we characterise further the important role of HuR in the entry and progression of GC B cells through the cell cycle. We have identified several important genes for the GC reaction- including Myc itself and Ccnd3 which is essential for GC cell cycling (Pae et al. J. Exp Med. 2021 Apr 5;218(4):e20201699, this paper is now cited in the manuscript) that are targets of HuR and deregulated in HuR cKO GC B cells. The molecular mechanisms by which HuR modulates the mRNA abundance and translation of these genes have been previously explored in different cell systems and are extensively discussed in our manuscript.

Specific points:

The data for a direct role of HuR in regulating Myc expression in GC B cells is quite weak. For data shown in Figure 4E, pre-gates would be helpful. The calculation of MFI on cells pre-gated on the same marker is

also not appropriate because this will lead to easily skewed data based on small numbers of outliers. For the MFI plot, the authors should show both of their datasets and use an appropriate test (data doesn't seem to be normally distributed) that takes into account their full dataset from both experiments.

Response – We have now revised the presentation of this data to present two independent experiments in Figure 4E. We have also re-analysed each experiment using a non-parametric Mann-Whitney U test that does not assume a normal distribution of the data. Additionally, we indicate in the figure legend that the gating strategy used for identification of Myc-expressing GC B cells is the one shown in Figure 3F.

Finally, to further substantiate that HuR is required for Myc expression in B cells, we have performed *in-vitro* B cell activation assays to assess whether Myc expression was decreased in HuR KO B cells compared to WT B cells. Although *in-vitro* stimulation does not recapitulate Tfh-dependent GC B cell activation, our experiments show that the induction of Myc in B cells requires of HuR expression. This is now presented in Figure 4F.

For dataset that combine data from multiple experiments where the experiment may introduce a shift in the overall level of the response (for example when showing both datasets in Figure 4E and possibly in Figure 7G) I would suggest the use of a linear mixed models with experiments as a blocking factor.

Response – For Figure 4E, we now present the data from the different experiments separately. Differences due to genotype are statistically significant in both replicates. In a similar manner, raw data plotted in Figure 7G collected from two independent experiments are presented as part of supplemental figure 7C. We believe this represents a good way to present the data for reader's evaluation. The impact of the batch effect variation does not diminish the finding of a difference.

The authors claim that HuR is dispensable for CSR and terminal diff into PC (lines 156) based on in vitro data. However, data in Figure 1C and 3A shows a much more pronounced effect on IgG1 ASCs (or GC B cells) than IgM+ cells. Given that CSR is happening independent of the GC (Roco et al, Immunity 2019) it seems likely that both processes are affected, at least for T-dependent responses.

Response – Our evidence supports the hypothesis that HuR is not essential for B-cell CSR and terminal differentiation. This is supported by *in-vitro* data after B cell activation with mitogens (Diaz-Munoz et al. Nat Immunol. 2015 Apr;16(4):415-25.) or after co-culture with 40LB stroma cells (Suppl. Figure 2A). Moreover, *in-vivo* analyses performed in this manuscript (Fig. 2B, Fig. 3A and Suppl. Fig 3) show that the reduction of IgG1+ GC B cells in the absence of HuR is dependent on the recognition of antigen. The number of IgG1+ GC B cells in emerging GCs (analysed 4.5 and 5.5 days after immunization, Suppl. Fig. 3C) shows no differences based on genotype. Differences in the percentage and number of IgG1+ GC B cells appears during the expansion and maintenance of the GC reaction. This is associated with antigen-mediated selection of B cell clones in GCs as the percentage of non-antigen binding IgG1+ GC B cells remain similar in control and HuR cKO mice (Fig. 3A). IgM antibody titers in serum and IgM ASCs remain unchanged when comparing control and HuR cKO mice. The percentage of IgM+ GC B cells increases during the progression of GC responses in these mice, most likely, due to impaired selection and the loss of antigen- specific class-switched GC B cells in the absence of HuR.

A key experimental mouse model for their paper is the use of AID-Cre mice. And while these mice have been widely used to delete genes selectively in GCs, the difference in cell number between mice using AID-Cre compared to Mb1-Cre (Supplementary Figure 1D) seems to indicate that the very early phase of the GC response (when HuR will be absent in Mb-1-cre mice but not necessarily in AID-Cre mice) seems to indicate a more fundamental role already in the very early establishment phase of the GC response.

Response – Initial experiments compared GC responses in AID-Cre, HuR^{fl/fl} and HuR^{fl/fl} AID-Cre mice. The number of GC B cells is reduced in HuR^{fl/fl} AID-Cre mice compared to either the AID-Cre or the HuR^{fl/fl} mice. Importantly, no differences were observed when comparing these two control groups. Thus, in most experiments we chose to compare littermates (HuR^{fl/fl} and HuR^{fl/fl} AID-Cre) to comply with good scientific practice and minimise use of mice (3Rs) while avoiding variation from the use of two different mouse lines (Holmdahl et al. EJI, 42, 1 2012). Results comparing GC responses in AID-Cre, HuR^{fl/fl} and HuR^{fl/fl} AID-Cre mice are now shown in the new Suppl. Fig. 1C. Finally, we previously reported that in HuR^{fl/fl} Mb1-Cre mice the establishment the extrafollicular plasmablast response and GC reaction is impaired as shown in Suppl. Fig. 1D and in Diaz-Munoz et al. Nat Immunol. 2015 Apr;16(4):415-25.

In Figure 1C the authors need to show the CD45.1/CD45.2 ratio in non-immunised control mice from the same experiment or the ratio in other cells within the same animal, for example T cells or non-GC B cells to demonstrate that the mice were reconstituted with a 1:3 ratio as stated. Statistical comparisons (as done in Figure 3D) should only be made between cell populations within the same animal to exclude unequal reconstitution effects between groups of mice receiving different donor BM mixes.

Response – We now show in Suppl. Fig. 3D the CD45.1/CD45.2 ratio of CD19+ IgD+ cell population from the same chimeras. This shows that the reconstitution (ratio) is equivalent between the two groups allowing us comparison between them.

What is the evidence for the claim (line 191) “HuR was efficiently deleted in over 90% of GC B cells”? Data in Suppl Figure 1b seems to show around 25% HuR+ GC B cells?

Response – We consistently observed efficient deletion of HuR using the AID-Cre mouse model. Deletion efficiency is 90% or higher at day 7 and 10 after immunization. This is shown in Fig. 3C in which HuR deletion is assessed in both control and HuR^{fl/fl} AID-Cre GC B cells in the same BM chimera mice. By contrast, HuR deletion is underestimated when assessed in HuR^{fl/fl} AID-Cre mice. This is most probably due to increased apoptosis and removal of HuR KO GC B cells that appears underrepresented compared to HuR-sufficient GC B cells (which are mostly IgM+). This becomes more apparent during the decline of GC responses at day 14 as shown in Fig. 3B.

Minor editorial points:

Number of NP-specific ASCs only shown for spleen, not BM (Fig 1C, line 134)

Response – This has been modified to remove BM from the text.

Can the authors please clarify the immunisation schedule? In some parts they write “immunised with NP-KLH alum for 7 days”. I assume this means analysis 7 days after the immunisation and not daily immunisation for 7 days?

Response – We have now revised the text to clearly state that most analyses were performed at day 7, 10 or 14 after immunisation.

Line 371: the 1.5-fold reduction seems very generously rounded. Based on the data in the Figure a 1.25x reduction seems more appropriate.

Response – This has been revised.

Line 375: Data seems to refer to Figure 7F and G.

Response – This has been revised.

Reviewer #3 (Remarks to the Author):

The manuscript "The RNA-binding protein HuR is required for maintenance of the germinal centre response" performs conditional deletion of the HuR to address the role of this RNA-binding protein in germinal centres. The study shows that HuR deficiency severely impairs the germinal centre reaction at different levels and through multiple mechanisms. The same authors had previously shown that HuR depletion at an earlier stage of B cell differentiation had a great impact on splicing, which led to defective mitochondrial metabolism and promoted B cell death. This new work provides a detailed and convincing characterization of the role of HuR in germinal centres. The results are novel and very interesting and the experimental approaches are appropriate. This is a solid piece of work.

Response – We thank the reviewer for his/her insightful comments and suggestions that have allowed us to improve our manuscript.

Comments and suggestions

1. One general comment on the manuscript regards the relative complexity of the phenotype observed and the molecular mechanisms proposed. The study unfolds the regulatory role of HuR on *myc*, but also suggests the direct regulation of *myc*-regulated genes by HuR. In addition, HuR regulates the abundance and splicing of numerous RNAs. The combination of these molecular events promotes, on one hand, a defect in the generation of high affinity antibodies (maybe in relation with *myc* and T cell help), and on the other, proliferation and possibly survival defects. A unifying model including these concepts would be very useful. Likewise, the discussion could expand a bit more on the interconnection between these different molecular mechanisms in the context of the germinal centre B cell (say, versus T-cell independent outcomes).

Response – We have now expanded the discussion as suggested to provide a unifying model as suggested (lines 516-525).

2. The study uses *HuR^{fl/fl}* and *HuR^{fl/fl} AID^{Cre}* as control and HuR deficient groups of mice, respectively. Ideally, both groups should carry the Cre allele to exclude rare lateral effects of Cre expression, thus in this case, the optimal setting would have been to compare *HuR^{+/+} AID^{Cre}* versus *HuR^{fl/fl} AID^{Cre}* mice. To dispel an effect of the Cre (and the AID transgene), it would be important to show some of the key findings (for instance, reduction of germinal centre size and high affinity antibodies) using *HuR^{+/+} AID^{Cre}* mice as controls.

Response – We compared GC responses in AID-Cre and *HuR^{fl/fl}* mice finding no differences between these two groups. Importantly, *HuR^{fl/fl}* AID-Cre mice showed a significant reduction in both the percentage and number of GC B cells compared to any of these two control groups that are equivalent. These data are now presented as part of Suppl. Fig. 1C. As noted above, having established that the AID cre-allele has no effect on the GC reaction. Like many colleagues (Holmdahl et al. EJI, 42, 1 2012), we have placed greater value on the use of littermates which share the same parents and environment (microbiota etc) from birth.

3. Figure 2 very nicely shows the germinal centre defect in HuR deficient mice. These results already suggest that proliferation (presumably in the DZ) and survival are defective. This could be directly addressed (here or later, when the cell cycle phenotype is further analyzed, at figure 7) with classical *in vitro* stimulation experiments (LPS, anti-CD40 IL4, etc). That would provide a more precise measure of cell

division (violet cell trace or CFSE) in combination with an analysis of isotype switch. The latter is also important in the context of the deamination signature and mutations shown in Figure 8. Also, the authors may want to more precisely measure cell death (caspase, tunnel).

Response – We and others have previously reported that *in-vitro* proliferation of HuR KO B cells as a measure of violet cell trace dilution and in response to mitogens is mildly affected compared to control cells (Diaz-Munoz et al. Nat Immunol. 2015 Apr;16(4):415-25. and DeMicco et al. J Immunol. 2015 Oct 1;195(7):3449-62). This is in contrast with our *in-vivo* results that suggest that HuR is required for the activation of a Myc-dependent transcriptional program and the cell cycle for the sustained expansion of antigen-specific GC B cells. Thus, the use of uridine analogues (and not of cell tracer dyes) was required to explore further the role of HuR in cell cycle progression *in-vivo* and *in-vitro*. Finally, we have now combined caspase and cell viability dyes to quantify cell death to reinforce the evidence that HuR prevents GC B cell death (new Fig 2E and Suppl. Fig. 2C).

4. In figure 5E, it would appear that no-HuR targets tend to be more highly expressed in KO DZ than in WT DZ (just the opposite as for HuR targets). If that is how the result is read, what is the interpretation?

Response – We prefer not to draw any conclusions regarding the no-HuR targets as we cannot assign these changes directly to the absence of HuR.

5. Figure 5 and 6 put show the effect of HuR defect in RNA abundance and in alternative splicing. Figure 6D and E connect these two mechanisms by showing common cellular pathways involved in both. However, it would be nice to have more information or discussion on the mechanistic connection as well. What is the overlap between these groups of genes? How can the effect of altering alternative splicing be predicted: can it lead (in some cases) to mRNA stability and thus lower abundance?

Response – We analysed whether alternative splicing (AS) in HuR KO GC B cells was linked to changes in mRNA abundance finding a little correlation between these molecular mechanisms (this is now explained in the main text, lanes 325-327). Briefly, we found that only 7 (out of 122) genes were both alternatively spliced and differentially expressed in HuR KO DZ cells compared to control cells. Similarly, only 14 out of 177 genes were both alternatively spliced and differentially expressed in HuR KO LZ cells compared to controls. The inclusion level of AS events associated to DE genes in HuR KO GC B cells was modest (around 10% or less) in general. However, changes in mRNA abundance was over 2- fold in most cases.

AS can also lead to aberrant mRNA translation in the presence or absence of significant changes in mRNA abundance. For example, *Elavl1* exon 2 is efficiently deleted (inclusion level = - 0.982) in HuR KO GC B cells, but *Elavl1* mRNA expression remains constant. By contrast, HuR controls the inclusion of exon 2 of *Ska2*, a protein required for chromosome segregation during mitosis (*Ska2* exon 2 inclusion level = - 0.64 in HuR KO GC B cells compared to control cells). *Ska2* mRNA abundance is reduced by over 1.5-fold in both DZ and LZ GC B cells in the absence of HuR. More importantly, prediction of mRNA translation suggests that exclusion of exon 2 leads to the synthesis of an alternative protein of 91 (instead of 120) amino acids with possibly altered function. Supplemental tables 1 and 2 contain extensive data regarding RNA abundance and alternative splicing, including genome annotation of all AS events and inclusion levels.

6. AS genes (Table S3) include *Aicda*, whose abundance is however not changed (Figure S7). There is a slight concern that transcripts arising from AIDCre transgenic construct could influence this AS finding. Could this be clarified?

Response – The transgenic construct (see Kwon et al. *Immunity* 28, 13 June 2008, Pages 751-762) is derived from a bacterial artificial chromosome in which Cre linked by an IRES to a truncated human CD2. The endogenous *Aicda* genes are not modified. Alternative splicing of *Aicda* affects exon 2 which appears to be skipped in HuR KO DZ and LZ B cells to a low degree (exclusion level of 0.109). So, it is likely that detection of *Aicda* alternative splicing comes from the AID-Cre transgenic construct. Nonetheless, this does not affect the interpretation of our results as *Aicda* mRNA abundance remains constant in control and HuR KO GC B cells. (This is now discussed in the manuscript, lanes 499-501).

7. Experiments on Figure S6 D-F are performed using mb1-Cre driver. Is there any reason for this? Classical in vitro experiments, ideally using HuR+/+ AIDCre versus HuR^{fl/fl} AIDCre mice would more precisely inform on cell division (please see points 2 and 3 above).

Response – We have performed *in-vitro* co-cultures of 40LB stroma cells and follicular B cells from HuR^{fl/fl} AID^{cre} mice. In this system, we observed a significant delay between the activation of the AID-Cre transgenic construct and effective Cre-mediated deletion of HuR. This limits the utility of the system *in vitro*. Thus, to precisely analyse the role of HuR in proliferation and the cell cycle we decided to use HuR KO FO B cells from HuR^{fl/fl} Mb1^{cre} mice.

8. The results in figure 8 are somehow difficult to interpret. Figure 8C shows a deaminase signature, but the clustering is not obvious: there seems to be disparity among the 3 replicates of each samples, and there is not a clear pattern across different samples. Can this be supported with statistical or PCA analysis?

Response – In Fig. 8C, hierarchical clustering is performed to assess sample distance similarity and formation of clusters based on genotype and cell type. This figure is not intended to show specific gene-associated patterns (although they can be observed for some genes including *Apobec1*, *Ada*, *Adal1* or *Dctd*). It supports the notion that global changes in the expression of genes associated with deamination are higher between WT DZ and LZ GC B cells. Furthermore, these changes are reduced in HuR KO cells as shown by the distance between KO DZ and LZ GC B cell clusters is smaller and between those clusters from WT GC B cells. Statistical analyses to test the significant of the changes associated to deaminase activity in HuR KO GC B cells compared to control cells are now shown in Fig. 8E.

Given that *Aicda* mRNA levels are not altered, how is the signature (ie other deaminases) relevant to the somatic hypermutation phenotype? Conversely, how can the increased somatic hypermutation be explained? Is this also accompanied by increased switch recombination? (please see point 3 above) Are the levels of AID protein increased?

Absence of changes in *Aicda* mRNA expression in HuR KO GC B cells cannot preclude changes in AID protein expression. We tried to assess AID protein expression by flow cytometry but we failed due to the lack of a specific antibody. We were also unable to perform more traditional analysis by immunoblotting due to the limited number of GC B cells that can be FAC-sorted from HuR^{fl/fl} AID^{cre} mice. Nonetheless, increased somatic hypermutation in HuR KO GC B cells as well as higher levels of pSer139-H2A.X suggest an increase in deaminase activity and DNA damage. This might be caused by *Aicda* or by *Apobec1*, which is significantly upregulated in HuR KO GC B cells. *Apobec1* was originally identified for its RNA editing capacity. However, many reports have now shown that *Apobec1* can act as a DNA mutator (Harris et al. *Mol Cell*. 10, 5, Nov 2002, Pages 1247-1253; Shivarov et al. *PNAS*, 2008 105 (41) 15866-15871; Saraconi et al. *Genome Biol* 15, 417 (2014).; Caval et al. *BMC Genomics* (2019) 20:858). Whether high expression of *Apobec1* in GC B cells leads to increased DNA mutagenesis and genotoxicity (associated or not to the Ig gene loci) remains to be tested.

Finally, we previously reported that class switch recombination was not increased in HuR KO B cells activated *in-vitro* with different mitogens (Diaz-Munoz et al. Nat Immunol. 2015 Apr;16(4):415-25.). In this manuscript we report that class switch recombination is not affected in B cells co-cultured with 40LB and in *in-vivo* assays showing that the percentage of no-binding antigen IgG1-class switched GC B cells is similar in control and HuR^{fl/fl} AID^{cre} mice.

Minor comments

1. Nomenclature of mouse strains used in the study is at times confusing. This is particularly so in Figure 4C. Does the top panel show HuR expression in immunized MycGFP/GFP? Figure legend is read somehow differently. Is the lower graph a quantification of the representative plots shown in the upper panel?

Response – This has now been reviewed and changed.

2. I may have missed this information: for the interpretation of the data in Figure S6B and C, it would be very useful to include as supplementary information the list of genes included in the G1 to S, S and G2 to M categories, as well as their expression in LZ and DZ cells.

Response – This has now been included in a new supplementary table.

3. Figure 6B. Please explain in figure legend the figures shown in the Venn diagram; possibly black refers to AS events and red refers to genes?

Response – Figure legend of Fig. 6B has now been revised to clarify this point.

4. Please show statistics for mutation analysis shown in Figure 8D-E.

Response – Statistics has now been added as requested.

REVIEWER COMMENTS

Reviewer #2 (Remarks to the Author):

I thank the authors for their thoughtful reply and apologise if some of my points were not clear. However, I maintain that the data presented in the original manuscript and the revised version does not address the key question whether HuR allows a Myc-dependent GC program or impairs the GC response independently of Myc and this defect in the GC response then results in a reduced Myc-dependent gene expression profile.

To reiterate and hopefully clarify, the authors convincingly show that absence of HuR has profound effects on the GC response and also that the absence of HuR has an effect on the set of Myc-regulated genes but no experiment convincingly links these two points in a way that would demonstrate that the effect on Myc-regulated genes impairs the GC response. And while the authors throughout the manuscript try to make the case that the effect on the Myc-regulated gene set is driving the failure of a GC response the opposite outcome (defect in GC response through an unknown mechanism causes a lack of Myc-driven typical GC gene expression changes) is at least as likely given that the enforced expression of Myc does not seem to rescue any phenotype. This strongly suggests that the key, physiologically relevant, role of HuR is either only on the downstream targets of Myc (but even these should be influenced by an enforced increase in Myc) or that HuR regulates some as yet unidentified genes independent of the Myc gene regulatory network. The view that HuR has a much broader role is also supported by the broad effect on alternative splicing and global changes to mRNA expression (Figure 5), suggesting a much broader role than only affecting a Myc-dependent program.

And in my view, this point is crucial to support statements like "HuR affects the transcriptome ... to enable the expression of a Myc-dependent transcriptional program ..." in the abstract and throughout the manuscript.

I apologise if my comment about the use of Mb1Cre vs AID-Cre mice and the data shown in Supplementary Figure 1 d/e (c/d in original submission) was not clear.

I fully support the use of littermates as controls over the use of animals from separate strains. My comment was about the apparent difference in the number and percentage of GC B cells between HuRfl/fl AIDCre and HuRfl/fl Mb1Cre mice at days 5.5 and 7.

This difference suggests that the absence of HuR during the initiation of the GC response (in Mb1Cre mice) but presence of HuR at the same stage in AIDCre mice (due to the time required for upregulation of AID-Cre, deletion of HuR and degradation of pre-existing HuR protein) has functional consequences. The importance of this point is also reinforced by the question (and answer) 7 of reviewer 3. This seems to make the statement that HuR plays a critical role in the expansion and/or maintenance but has no effect on early phase far too strong.

Furthermore, I would strongly suggest to re-calculate the significance values for the day 7 data in panels d and E, given the overlap in populations between the HuRfl/fl and HuRfl/fl AIDCre mice the values seem incredibly low.

Reviewer #3:

These comments have been mediated by reviewer #2. Their assessment of how the comments have been addressed are noted in "".

Remarks to the Author:

The manuscript "The RNA-binding protein HuR is required for maintenance of the germinal centre response" performs conditional deletion of the HuR to address the role of this RNA-binding protein in germinal centres. The study shows that HuR deficiency severely impairs the germinal centre reaction at different levels and through multiple mechanisms. The same authors had previously shown that HuR depletion at an earlier stage of B cell differentiation had a great impact on splicing, which led to defective mitochondrial metabolism and promoted B cell death. This new work provides a detailed and convincing characterization of the role of HuR in germinal centres. The results are novel and very interesting and the experimental approaches are appropriate. This is a solid piece of work.

Comments and suggestions

1. One general comment on the manuscript regards the relative complexity of the phenotype observed and the molecular mechanisms proposed. The study unfolds the regulatory role of HuR on *myc*, but also suggests the direct regulation of *myc*-regulated genes by HuR. In addition, HuR regulates the abundance and splicing of numerous RNAs. The combination of these molecular events promotes, on one hand, a defect in the generation of high affinity antibodies (maybe in relation with *myc* and T cell help), and on the other, proliferation and possibly survival defects. A unifying model including these concepts would be very useful. Likewise, the discussion could expand a bit more on the interconnection between these different molecular mechanisms in the context of the germinal centre B cell (say, versus T-cell independent outcomes).

"They have provided a unifying model in agreement with their interpretation of the data. As indicated in my own comments, I don't fully agree with their interpretation but the answer is definitely a clear statement of their interpretation and assumptions."

2. The study uses HuR^{fl/fl} and HuR^{fl/fl} AID^{Cre} as control and HuR deficient groups of mice, respectively. Ideally, both groups should carry the Cre allele to exclude rare lateral effects of Cre expression, thus in this case, the optimal setting would have been to compare HuR^{+/+} AID^{Cre} versus HuR^{fl/fl} AID^{Cre} mice. To dispel an effect of the Cre (and the AID transgene), it would be important to show some of the key findings (for instance, reduction of germinal centre size and high affinity antibodies) using HuR^{+/+} AID^{Cre} mice as controls.

"Fully addressed."

3. Figure 2 very nicely shows the germinal centre defect in HuR deficient mice. These results already suggest that proliferation (presumably in the DZ) and survival are defective. This could be directly addressed (here or later, when the cell cycle phenotype is further analyzed, at figure 7) with classical in vitro stimulation experiments (LPS, anti-CD40 IL4, etc). That would provide a more precise measure of cell division (violet cell trace or CFSE) in combination with an analysis of isotype switch. The latter is also important in the context of the deamination signature and mutations shown in Figure 8. Also, the authors may want to more precisely measure cell death (caspase, tunnel).

"While not every suggested experiment was performed, overall the comments have been addressed to some extend."

4. In figure 5E, it would appear that no-HuR targets tend to be more highly expressed in KO DZ than in WT DZ (just the opposite as for HuR targets). If that is how the result is read, what is the interpretation?

"While I understand the reluctance of the authors to draw too many conclusions from the non-HuR targets the reviewer has raised an important point. In relation to that, if I understand the figure correctly, the statistical test compares non-HuR targets with HuR targets and shows that there is a statistically significant difference. However, the test does not say if the difference comes from a reduced expression of the HuR targets (as emphasised by the authors) or from an increased expression of non-HuR target genes. Because of that I think a comment on the non-HuR targets is warranted and/or a different comparison to be used to show that HuR targets have reduced expression."

5. Figure 5 and 6 put show the effect of HuR defect in RNA abundance and in alternative splicing. Figure 6D and E connect these two mechanisms by showing common cellular pathways involved in both. However, it would be nice to have more information or discussion on the mechanistic connection as well. What is the overlap between these groups of genes? How can the effect of altering alternative splicing be predicted: can it lead (in some cases) to mRNA stability and thus lower abundance?

"Comment is addressed."

6. AS genes (Table S3) include *Aicda*, whose abundance is however not changed (Figure S7). There is a slight concern that transcripts arising from AIDCre transgenic construct could influence this AS finding. Could this be clarified?

"Addressed."

7. Experiments on Figure S6 D-F are performed using mb1-Cre driver. Is there any reason for this? Classical in vitro experiments, ideally using HuR^{+/+} AIDCre versus HuR^{fl/fl} AIDCre mice would more precisely inform on cell division (please see points 2 and 3 above).

"Somewhat addressed but the answer here about a delayed deletion in AIDCre mice goes to the point that I raised in my review and that was not addressed or misunderstood."

8. The results in figure 8 are somehow difficult to interpret. Figure 8C shows a deaminase signature, but the clustering is not obvious: there seems to be disparity among the 3 replicates of each samples, and there is not a clear pattern across different samples. Can this be supported with statistical or PCA analysis? Given that *Aicda* mRNA levels are not altered, how is the signature (ie other deaminases) relevant to the somatic hypermutation phenotype? Conversely, how can the increased somatic hypermutation be explained? Is this also accompanied by increased switch recombination? (please see point 3 above) Are the levels of AID protein increased?

"I Agree with the reviewer that this figure was hard to interpret and I don't think this questions was addressed. They have added a statistical comparison for the mutation rate in JH4 but this seems to be in response to the "Minor comment 4" and not as an answers to the question about statistical/PCA analysis for the deaminase signature raised by reviewer 3."

Minor comments

"All minor comments were addressed"

1. Nomenclature of mouse strains used in the study is at times confusing. This is particularly so in Figure 4C. Does the top panel show HuR expression in immunized MycGFP/GFP? Figure legend is read somehow differently. Is the lower graph a quantification of the representative plots shown in the upper panel?

2. I may have missed this information: for the interpretation of the data in Figure S6B and C, it would be very useful to include as supplementary information the list of genes included in the G1 to S, S and G2 to M categories, as well as their expression in LZ and DZ cells.

3. Figure 6B. Please explain in figure legend the figures shown in the Venn diagram; possibly black refers to AS events and red refers to genes?

4. Please show statistics for mutation analysis shown in Figure 8D-E.

Point-by-point response to reviewers

We thank the reviewer for his insightful comments and hope that our responses have improved the manuscript so that it is now worthy of publication.

Reviewer #2 (Remarks to the Author):

I thank the authors for their thoughtful reply and apologise if some of my points were not clear. However, I maintain that the data presented in the original manuscript and the revised version does not address the key question whether HuR allows a Myc-dependent GC program or impairs the GC response independently of Myc and this defect in the GC response then results in a reduced Myc-dependent gene expression profile.

Response –HuR plays pleiotropic functions in GC B cells, controlling not only the expression of Myc and Myc-dependent genes but also the mRNA splicing and expression of genes associated to cell cycle and DNA damage.

We cannot rule out completely that other mechanism regulated by HuR (e.g. proliferation and DNA damage) have an impact on later LZ B cell selection and expression of a Myc-dependent program. Nonetheless, transcriptomics analyses in HuR KO GC B cells did not reveal any defects in relevant signalling pathways activated during Tfh cell-mediated selection and required for Myc induction (e.g. PI3K or NF- κ B pathways; Sander *et al. Immunity* 2015; Heise *et al. JEM* 2014; Lue *et al. Immunity* 2018). BCR- expression in the cell surface was also similar in WT and HuR KO GC B cells (**see page 14, line 447-449**). Additional *in-vitro* analyses did not show alterations in the release of Ca²⁺ or activation of HuR KO B cells upon stimulation through the BCR or the CD40 receptor (Diaz-Munoz *et al. Nat. Immunol.* 2015) which suggest that HuR is dispensable for signal transduction upon activation.

By contrast, we show selective binding of HuR to the mRNA of Myc and Myc-associated genes and a global downregulation of these genes in the absence of HuR (**figure 4**). Further evidences from an independent group have also highlighted that post-transcriptional modification and stabilization of Myc mRNA is an active mechanism for Myc protein synthesis in LZ GC B cells and, if altered, can compromise GC responses (Grenov *et al. <https://doi.org/10.1101/2020.09.08.287433>*). Taken together, these evidences indicate that HuR contributes directly to the expression of Myc and of a Myc-dependent program in GC B cells. This, along with further roles of HuR in the control of the cell cycle, the DNA damage response and possibly other pathways, will explain the defects observed in the development and maintenance of GC responses in HuR^{fl/fl} AID^{Cre} mice.

To reiterate and hopefully clarify, the authors convincingly show that absence of HuR has profound effects on the GC response and also that the absence of HuR has an effect on the set of Myc-regulated genes but no experiment convincingly links these two points in a way that would demonstrate that the effect on Myc-regulated genes impairs the GC response.

Response – Data presented in **figure 4d, e and f** and discussed in **page 8-9, lines 253-269** highlight that HuR binds to the 3'UTR of Myc and is required for the full induction of Myc in both positively selected LZ GC B cells and in B cells activated *in-vitro* with different mitogens (LPS, anti-IgM or anti-CD40). This latest result is especially relevant as reveals a direct implication of HuR in the expression of Myc that is independent of the type of stimulation received by B cells. Several reports have now linked dynamic regulation of c-Myc

expression (both in time and magnitude) with the launch of a Myc-dependent program that sets the fate of GC B cells (Dominguez-Sola et al. *Nat Immunol* 2021; Calado et al. *Nat. Immunol.* 2012; Finklin et al. *Immunity* 2019; Nakagawa et al. *PNAS* 2021).

And while the authors throughout the manuscript try to make the case that the effect on the Myc-regulated gene set is driving the failure of a GC response the opposite outcome (defect in GC response through an unknown mechanism causes a lack of Myc-driven typical GC gene expression changes) is at least as likely given that the enforced expression of Myc does not seem to rescue any phenotype. This strongly suggests that the key, physiologically relevant, role of HuR is either only on the downstream targets of Myc (but even these should be influenced by an enforced increase in Myc) or that HuR regulates some as yet unidentified genes independent of the Myc gene regulatory network.

Response – Dysregulation of Myc contributes to impaired GC responses in HuR cKO mice although it is unlikely to be the only event that cause the phenotype as discussed extensively in the manuscript. We have identified key roles for HuR in the regulation of cell cycle and DNA damage that, if these are altered, can compromise GC responses in a Myc-independent manner (*see discussion*). It is thus unsurprising that enforced expression of Myc does not rescues GC responses in HuR^{fl/fl} AID^{Cre} mice. HuR regulates the mRNA abundance of Myc-dependent genes at the post-transcriptional level (*figure 7b*). Thus, it is very likely that gene transcriptional activation upon enforced expression of Myc gets compromised by the lack of HuR-dependent stabilization of newly transcribed mRNAs (*see discussion*).

The view that HuR has a much broader role is also supported by the broad effect on alternative splicing and global changes to mRNA expression (Figure 5), suggesting a much broader role than only affecting a Myc-dependent program.

Response – We agree with the reviewer that post-transcriptional regulation by HuR has broad implications for gene expression in GC B cells. A key aspect to elucidate in the future is whether altered mRNA splicing in HuR KO GC B cells leads to differential protein isoform expression or a reduced synthesis of key metabolic proteins. HuR has also been involved in timely mRNA transcription. Our current efforts aim to characterise the transcriptome of WT and HuR KO B cells and integrate this with our transcriptomics and HuR:RNA interactome analyses. This approach will likely identify further differences in HuR KO GC B cells associated to alternative mRNA splicing or translation that are not necessary reflected at the mRNA level.

And in my view, this point is crucial to support statements like “HuR affects the transcriptome ... to enable the expression of a Myc-dependent transcriptional program ...” in the abstract and throughout the manuscript.

Response – We have modified the abstract and the manuscript to reiterate that HuR is required for the expression of hundreds of genes including Myc. We extensively discuss in *pages 15* other important implications of HuR in the expression of genes associated to cell cycle and proliferation. These programs will certainly have an impact on GC expansion and maintenance by their own. Our HuR:RNA interactome and transcriptome data shows that HuR binds to the 3'UTR and regulate the expression of key cyclins in the proliferation of eukaryotic cells including GC B cells (e.g. Ccnd2, Ccnd3, Ccnb1) (Pae et al., 2021; W. Wang et al., 2000). Finally, we discuss the importance of timely control of genotoxic stress responses in germinal centres in *page 15-16*.

I apologise if my comment about the use of Mb1Cre vs AID-Cre mice and the data shown in Supplementary Figure 1 d/e (c/d in original submission) was not clear. I fully support the use of littermates as controls over the use of animals from separate strains. My comment was about the apparent difference in the number and percentage of GC B cells between HuR^{fl/fl} AIDCre and HuR^{fl/fl} Mb1Cre mice at days 5.5 and 7. This difference suggests that the absence of HuR during the initiation of the GC response (in Mb1Cre mice) but presence of HuR at the same stage in AIDCre mice (due to the time required for upregulation of AID-Cre, deletion of HuR and degradation of pre-existing HuR protein) has functional consequences. The importance of this point is also reinforced by the question (and answer) 7 of reviewer 3. This seems to make the statement that HuR plays a critical role in the expansion and/or maintenance but has no effect on early phase far too strong.

Response – We thank the reviewer for raising the matter regarding the differences in the establishment of GCs in HuR^{fl/fl} AID^{Cre} and HuR^{fl/fl} Mb1^{Cre} mice and apologise for the little explanation provided to this question in our original manuscript. We have now experimentally reproduced our findings and provided an extended discussion of the results (**page 5-6, lines 148-161**).

Early GC responses at day 4.5 and 5.5 post-immunization are impaired in HuR^{fl/fl} Mb1^{Cre} mice but not in HuR^{fl/fl} AID^{Cre} mice (**Supplementary figure 1d and 1e**). The failure of HuR^{fl/fl} Mb1^{Cre} mice to mount efficient GC responses most likely reflects the requirement of HuR expression for *in-vivo* activation of the B cell metabolism upon antigen encounter. We have previously reported that both T-dependent and T-independent immune responses are defective in HuR^{fl/fl} Mb1^{Cre} mice (Diaz-Munoz et al. *Nat. Immunol.* 2015). We have also showed that HuR is required upon activation for the early B-cell metabolic switch that provides energy and protects B cells from the cytotoxic effects of exacerbated production of reactive oxygen species. Further observations (*unpublished*) revealed that HuR KO B cells present a competitive disadvantage in the presence of WT B cells. Only HuR-sufficient B cells are found in GCs of mixed-bone marrow chimera mice or HuR^{fl/fl} CD19^{Cre} mice, in which we have estimated that around 10% of FO B cells escape from Cre-mediated recombination. Although we cannot discard that HuR plays an important intrinsic role in B cell commitment into GC B cells, our data suggest that this putative role is likely to be masked by the key implication of HuR in early B cell metabolic activation and proliferation. This limits the usefulness of HuR^{fl/fl} Mb1^{Cre} mice as a model to study GC responses and impedes us for making strong claims about the implication of HuR in GC formation.

Regarding the results from HuR^{fl/fl} AID^{Cre} mice, we agree with the interpretation of the reviewer. Differences found in early GC responses in HuR^{fl/fl} Mb1^{Cre} and HuR^{fl/fl} AID^{Cre} mice are likely to be due to the time required for transcriptional activation of the AID-Cre transgenic construct, Cre protein expression and efficient depletion of HuR in GC B cells when using the HuR^{fl/fl} AID^{Cre} mouse model. Recent evidences by Roco et al. (*Immunity*, Aug 2019) showed that the *Aicda* endogenous gene locus is transcriptionally active at day 2.5 post-immunization. However, transcription from the AID-Cre transgene is not taking place until day 4.5 as shown by the analysis of the hCD2 reporter (**Supplementary figure 1d**).

Furthermore, I would strongly suggest to re-calculate the significance values for the day 7 data in panels d and e, given the overlap in populations between the HuRfl/fl and HuRfl/fl AIDcre mice the values seem incredibly low.

Response – We have now reanalysed old and new data regarding the establishment of GCs in HuR^{fl/fl}, HuR^{fl/fl} AID^{Cre} and HuR^{fl/fl} Mb1^{Cre} (presented in new **Supplementary Figure 1d and 1e**). Data from three independent experiments are now plotted in a linear scale for better visualization (without a break in the scale as in the previous version). As the percentage and the number of GC B cells (gated as CD19⁺ IgD⁻ CD95⁺) is very low at

day 3.5 and day 4.5 post-immunization, a new plot is provided to show the data using a more suitable y-axis scale.

Reviewer #3:

These comments have been mediated by reviewer #2. Their assessment of how the comments have been addressed are noted in "".

Original Remarks to the Author from reviewer #3:

The manuscript "The RNA-binding protein HuR is required for maintenance of the germinal centre response" performs conditional deletion of the HuR to address the role of this RNA-binding protein in germinal centres. The study shows that HuR deficiency severely impairs the germinal centre reaction at different levels and through multiple mechanisms. The same authors had previously shown that HuR depletion at an earlier stage of B cell differentiation had a great impact on splicing, which led to defective mitochondrial metabolism and promoted B cell death. This new work provides a detailed and convincing characterization of the role of HuR in germinal centres. The results are novel and very interesting and the experimental approaches are appropriate. This is a solid piece of work.

Comments and suggestions

1. One general comment on the manuscript regards the relative complexity of the phenotype observed and the molecular mechanisms proposed. The study unfolds the regulatory role of HuR on *myc*, but also suggests the direct regulation of *myc*-regulated genes by HuR. In addition, HuR regulates the abundance and splicing of numerous RNAs. The combination of these molecular events promotes, on one hand, a defect in the generation of high affinity antibodies (maybe in relation with *myc* and T cell help), and on the other, proliferation and possibly survival defects. A unifying model including these concepts would be very useful. Likewise, the discussion could expand a bit more on the interconnection between these different molecular mechanisms in the context of the germinal centre B cell (say, versus T-cell independent outcomes).

Assessment to our response from reviewer #2: *"They have provided a unifying model in agreement with their interpretation of the data. As indicated in my own comments, I don't fully agree with their interpretation but the answer is definitely a clear statement of their interpretation and assumptions."*

Response – We thank reviewer 2 for taking on board the task of assessing our responses to reviewer 3. We have now reviewed our proposed model to highlight that HuR is likely to have ample implications in the regulation of the transcriptome (by controlling mRNA splicing, stability and/or translation). This includes, but not exclusively, the regulation of *Myc*, *Myc*-dependent genes and genes associated to the cell cycle (**page 16, lines 527-528**).

2. The study uses *HuR^{fl/fl}* and *HuR^{fl/fl} AID^{Cre}* as control and *HuR* deficient groups of mice, respectively. Ideally, both groups should carry the *Cre* allele to exclude rare lateral effects of *Cre* expression, thus in this case, the optimal setting would have been to compare *HuR^{+/+} AID^{Cre}* versus *HuR^{fl/fl} AID^{Cre}* mice. To dispel an effect of the *Cre* (and the *AID* transgene), it would be important to show some of the key findings (for instance, reduction of germinal centre size and high affinity antibodies) using *HuR^{+/+} AID^{Cre}* mice as controls.

Assessment to our response from reviewer #2: *"Fully addressed."*

3. Figure 2 very nicely shows the germinal centre defect in HuR deficient mice. These results already suggest that proliferation (presumably in the DZ) and survival are defective. This could be directly addressed (here or later, when the cell cycle phenotype is further analyzed, at figure 7) with classical in vitro stimulation experiments (LPS, anti-CD40 IL4, etc). That would provide a more precise measure of cell division (violet cell trace or CFSE) in combination with an analysis of isotype switch. The latter is also important in the context of the deamination signature and mutations shown in Figure 8. Also, the authors may want to more precisely measure cell death (caspase, tunnel).

Assessment to our response from reviewer #2: *"While not every suggested experiment was performed, overall the comments have been addressed to some extent."*

Response – Reviewer 3 proposed classical in-vitro B-cell activation assays to track cell proliferation by dilution of a cell tracer dye. Mitogens used in such experiments (LPS, anti-CD40+IL4) does not mimic the activation and proliferation of GC B cells. Instead we have assessed B cell proliferation, Ig-class switch recombination and cell cycle progression in B cells cocultured with 40LB stroma cells. Although this *in-vitro* system does not fully recapitulate the events in GCs, it generates GC-like B cells that upregulate the GC B cell markers CD95 and GL7.

Finally, we decided not perform tunnel assays as suggested by reviewer 3. This assay uses a terminal deoxynucleotide transferase to efficiently label DNA breaks occurring at late stages during apoptosis. Most healthy cells efficiently repair DNA breaks and are negative in these tunnel assays. However, in GC B cells expressing AID, naturally occurring DNA breaks and mutagenesis impedes measuring apoptosis with a tunnel assay. Instead we combined detection of cell permeabilization and detection of intracellular primary amine groups on proteins (Fixable viability dyes) with the detection of the cell-permeable inhibitor FITC-VAD-FMK that binds irreversibly to the catalytic active site of caspases in apoptotic cells (**Figure 2e**).

4. In figure 5E, it would appear that no-HuR targets tend to be more highly expressed in KO DZ than in WT DZ (just the opposite as for HuR targets). If that is how the result is read, what is the interpretation?

Assessment to our response from reviewer #2: *"While I understand the reluctance of the authors to draw too many conclusions from the non-HuR targets the reviewer has raised an important point. In relation to that, if I understand the figure correctly, the statistical test compares non-HuR targets with HuR targets and shows that there is a statistically significant difference. However, the test does not say if the difference comes from a reduced expression of the HuR targets (as emphasised by the authors) or from an increased expression of non-HuR target genes. Because of that I think a comment on the non-HuR targets is warranted and/or a different comparison to be used to show that HuR targets have reduced expression."*

Response – Our aim in figure 5e is to show the directionality in the expression of differentially expressed genes in HuR KO GC B cells based on whether their mRNA transcripts are bound or not with HuR. This is indeed the case for HuR KO DZ B cells, with a Kolmogorov-Smirnov test (to assess whether the distribution of the data differs between HuR targets and non-HuR targets) and a Wilcox test (for comparison of the means) showing significance ($p=0.0107$ and $p=0.0073$, respectively). Thus, we have now modified **figure 5e** and the text (**page 10, lines 310-313**) to indicate that 21 out of 39 DE genes targeted by HuR were found downregulated in HuR KO DZ B cells. By contrast, 36 out of 51 non HuR targeted DE genes were increased in HuR KO DZ B cells. This highlights that the difference in the distribution of the data in figure 5e (left panel) are likely due to both effects.

Unfortunately, our data does not allow us to draw any conclusions about non-HuR targets, which expression most likely changes due to regulatory mechanisms that are not directly linked with HuR. By contrast, we could test whether the overall expression of HuR gene targets was affected in HuR KO GC B cells. This analysis (presented in figure 5d and supplementary figure 5) reveals that in a genome-wide scale HuR contributes to the maintenance of the mRNA abundance of its targets.

5. Figure 5 and 6 put show the effect of HuR defect in RNA abundance and in alternative splicing. Figure 6D and E connect these two mechanisms by showing common cellular pathways involved in both. However, it would be nice to have more information or discussion on the mechanistic connection as well. What is the overlap between these groups of genes? How can the effect of altering alternative splicing be predicted: can it lead (in some cases) to mRNA stability and thus lower abundance?

Assessment to our response from reviewer #2: *"Comment is addressed."*

6. AS genes (Table S3) include Aicda, whose abundance is however not changed (Figure S7). There is a slight concern that transcripts arising from AIDCre transgenic construct could influence this AS finding. Could this be clarified?

Assessment to our response from reviewer #2: *"Addressed."*

7. Experiments on Figure S6 D-F are performed using mb1-Cre driver. Is there any reason for this? Classical in vitro experiments, ideally using HuR+/+ AIDCre versus HuRfl/fl AIDCre mice would more precisely inform on cell division (please see points 2 and 3 above).

Assessment to our response from reviewer #2: *"Somewhat addressed but the answer here about a delayed deletion in AIDCre mice goes to the point that I raised in my review and that was not addressed or misunderstood."*

Response – This point is addressed above in response to the specific comment from reviewer 2.

8. The results in figure 8 are somehow difficult to interpret. Figure 8C shows a deaminase signature, but the clustering is not obvious: there seems to be disparity among the 3 replicates of each samples, and there is not a clear pattern across different samples. Can this be supported with statistical or PCA analysis? Given that Aicda mRNA levels are not altered, how is the signature (ie other deaminases) relevant to the somatic hypermutation phenotype? Conversely, how can the increased somatic hypermutation be explained? Is this also accompanied by increased switch recombination? (please see point 3 above) Are the levels of AID protein increased?

Assessment to our response from reviewer #2: *"I Agree with the reviewer that this figure was hard to interpret and I don't think this question was addressed. They have added a statistical comparison for the mutation rate in JH4 but this seems to be in response to the "Minor comment 4" and not as an answer to the question about statistical/PCA analysis for the deaminase signature raised by reviewer 3."*

Response – We thank the reviewers for pointing out the difficulties encountered in the visualization and interpretation of the results from our gene set enrichment analysis (GSEA) that revealed an increased expression of genes associated to deamination in HuR KO GC B cells. We have revised the data presented in **figure 8b and 8c** and show now the full statistics from the GSEA for DZ and LZ GC B cells (gene enrichment score (ES), normalized enrichment score (NES) and false discovery rate (FDR) as well as indicate which are the leading-edge genes driving pathway upregulation in HuR KO GC B cells. Finally, we decided to show the relative expression of genes involved in deaminase activity in DZ and LZ GC B cells separately to simplify visualization and interpretation of the data. Text associated to figure 8 b and 8c has been modified accordingly (**page 13, lines 411-412 and page 32, lines 1069-1075**). All other questions from reviewer 3 were responded in our previous rebuttal letter.

Minor comments

- 1. Nomenclature of mouse strains used in the study is at times confusing. This is particularly so in Figure 4C. Does the top panel show HuR expression in immunized MycGFP/GFP? Figure legend is read somehow differently. Is the lower graph a quantification of the representative plots shown in the upper panel?**
- 2. I may have missed this information: for the interpretation of the data in Figure S6B and C, it would be very useful to include as supplementary information the list of genes included in the G1 to S, S and G2 to M categories, as well as their expression in LZ and DZ cells.**
- 3. Figure 6B. Please explain in figure legend the figures shown in the Venn diagram; possibly black refers to AS events and red refers to genes?**
- 4. Please show statistics for mutation analysis shown in Figure 8D-E.**

Assessment to our response from reviewer #2: *"All minor comments were addressed"*

Response – We thank again to reviewer 2 for helping out with assessing our responses to reviewer 3. We sincerely appreciated the time dedicated to improve our manuscript.

REVIEWERS' COMMENTS

Reviewer #2 (Remarks to the Author):

I thank the authors for the careful consideration of my comments and hope that they contributed to improve the manuscript. I apologise for the delay in reviewing the final submission.

REVIEWERS' COMMENTS

Reviewer #2 (Remarks to the Author):

I thank the authors for the careful consideration of my comments and hope that they contributed to improve the manuscript. I apologise for the delay in reviewing the final submission.

We thank the reviewer for the help provided. His comments and suggestions contributed greatly to improve our manuscript.

Reviewer #2 comments to editor about mediating reviewer #3 comments:

I have looked again at the manuscript and rebuttal letter and all reviewer 3 comments have been addressed.

We thank reviewer 2 for the help provided assessing our response to reviewer 3 comments. His suggestions have contributed greatly to improve our manuscript.